



# Spatial and temporal variations in basal melting at Nivlisen ice shelf, East Antarctica, derived from phase-sensitive radars

**Katrin Lindbäck[1], Geir Moholdt[1], Keith W. Nicholls[2], Tore Hattermann[1], Bhanu Pratap[3], Meloth Thamban[3], Kenichi Matsuoka[1]**

[1] Norwegian Polar Institute, Framsentret, Postboks 6606, Langnes, 9296 Tromsø, Norway.

[2] British Antarctic Survey, Natural Environmental Research Council, High Cross, Madingley Rd, Cambridge CB3 0ET, UK.

[3] ESSO-National Centre for Polar and Ocean Research, Ministry of Earth Sciences, Headland Sada, Vasco-da-Gama, Goa 403 804 , India.

*Correspondence to:* Katrin Lindbäck (katrin.lindback@npolar.no)

**Abstract**

Thinning rates of ice shelves vary widely around Antarctica and basal melting is a major component in ice shelf mass loss. In this study, we present records of basal melting, at unique spatial and temporal resolution for East Antarctica, derived from autonomous phase-sensitive radars. These records show spatial and temporal variations of ice shelf basal melting in 2017 and 2018 at Nivlisen, central Dronning Maud Land. The annually averaged melt rates are in general moderate (~0.8 m yr$^{-1}$). Radar profiling of the ice-shelf shows variable ice thickness from smooth beds to basal crevasses and channels. The highest melt rates (3.9 m yr$^{-1}$) were observed close to a grounded feature near the ice shelf front. Daily time-varying measurements reveal a seasonal melt signal 4 km from the ice shelf front, at an ice draft of 130 m, where the highest daily melt rates occurred in summer (up to 5.6 m yr$^{-1}$). This seasonality indicates that summer-warmed ocean surface water was pushed by wind beneath the ice shelf front. We observed a different melt regime 35 km into the ice-shelf cavity, at an ice draft of 280 m, with considerably lower melt rates (annual average of 0.4 m yr$^{-1}$) and no seasonality. We conclude that warm deep





ocean water at present has limited effect on the basal melting of Nivlisen. On the other hand, a
warming in surface waters, as a result of diminishing sea-ice cover has the potential to increase
basal melting near the ice-shelf front. Many ice shelves like Nivlisen are stabilized by pinning
points at their ice fronts and these areas may be vulnerable to future change.

## 32    1   Introduction

The Antarctic contribution to global sea-level rise has increased by a factor of five in the

past two decades (The IMBIE Team, 2018). This rapid increase in the overall mass deficit is
mostly caused by several retreating and thinning glaciers in West Antarctica that lost buttressing
forces from their shrinking ice shelves (De Angelis and Skvarca, 2003; Joughin et al., 2014;
Rignot et al., 2014). Over 80 % of the grounded ice in Antarctica drains out into floating ice
shelves (Dupont and Alley, 2005). The thinning rates of these ice shelves vary widely around the
continent (Paolo et al., 2015). The mass balance of an ice shelf is the sum of the ice gain and
loss; ice gain comprises the advective input of grounded ice upstream, snow accumulation, and
marine ice accretion, and ice loss encompasses basal melting from the ocean and iceberg calving
(Bamber et al., 2018). A negative mass balance can affect ice-shelf stability: where the mass loss
reduces back stress on grounded ice upstream of the ice shelf, leading them to flow faster (Reese
et al., 2018). Understanding controls on the mass balance of ice shelves around Antarctica is
therefore key to gaining a better understanding of the continent's present and future contribution
to global sea-level rise.

Iceberg calving occurs irregularly in time and can have dramatic effects on ice shelf mass

balance when it occurs (Hogg and Gudmundsson, 2017). At present, however, basal melting is
the largest mass-loss process for Antarctic ice shelves (Depoorter et al., 2013; Rignot et al.,
2013). Melting of ice shelves by the ocean is not uniform and depends on the ocean properties in
the vicinity of the ice shelf, the topography of both the ocean bed and the ice-shelf base. Jacobs
et al. (1992) described three different modes of melting: In mode 1, ocean water with
temperatures at the surface freezing point provides heat for basal melting of deeper parts of the
ice base, because the pressure-melting point of the ice is decreased to lower temperatures at
depth. Since these cold shelf waters provide a limited source of ocean heat (Darelius and Sallée,



2017), average melt rates are often low for the largest ice shelves (e.g., 0.3 m yr$^{-1}$ for Ronne Ice
Shelf with; Rignot et al., 2013). In addition, substantial marine ice accretion occurs when the
rising melt plume from the grounding zone super-cools and refreezes on the ice-shelf base at
shallower depths (Joughin and Vaughan, 2004).
The rapid retreat and high thinning rates of glaciers in the Amundsen Sea sector of West
Antarctica are thought to be driven by an increased presence of warm circumpolar deep water
below the ice shelves (Pritchard et al., 2012; Rignot et al., 2013), referred to as melt mode 2 in
Jacobs et al. (1992). Circumpolar deep water surrounds the Antarctic continent, flowing
clockwise with the Antarctic Circumpolar Current and is abundant near the continental shelf of
West Antarctica. Circumpolar deep water accesses the deep bases of ice shelves directly through
cross-continental submarine troughs, causing high melt rates; for example Rignot et al. (2013)
found Pine Island Ice Shelf to have an average melt rate of 16 m yr$^{-1}$. In East Antarctica, basal
melting has been linked to circumpolar deep water intrusion only at Tottem Ice Shelf, where
annual basal melt rates reached ~11 m yr$^{-1}$ (Rignot et al., 2013; Rintoul et al., 2016). Farther
west, in the Weddell Sea sector a cooler modified version of circumpolar deep water is advected
along the coast (Dong et al., 2016; Ryan et al., 2016).
Ice shelves can also melt in the vicinity of their ice fronts when summer-warmed
Antarctic surface water is pushed by wind and tides under ice shelves (Jenkins and Doake, 1991;
Makinson and Nicholls, 1999; Sverdrup, 1954; Zhou et al., 2014). Jacobs et al. (1992) refer to
this as melt mode 3. Antarctic surface water has only recently been observed at Ross Ice Shelf
(Malyarenko et al., 2019; Stern et al., 2013; Stewart et al., 2019) and at Fimbulisen (Hattermann
et al., 2012), suggesting it may be an important process in basal melting. Spatial patterns and
relative magnitudes of all these three modes remain largely unknown. Numerical modelling,
however, indicates that the response of basal melting in the future strongly depends on the
surface air warming (Kusahara and Hasumi, 2013). Future basal melting in Antarctica will
therefore reflect the integrated response to remotely changes in sub-surface circumpolar deep
water temperatures and the coastal processes that control its access to the continental shelf
(Thompson et al., 2018) and the local upper ocean heat supply. The detailed interplay of these
processes today and in a future climate are still a major source of uncertainty when evaluating
the response of the Antarctic Ice Sheet to climate change (Adusumilli et al., 2018).





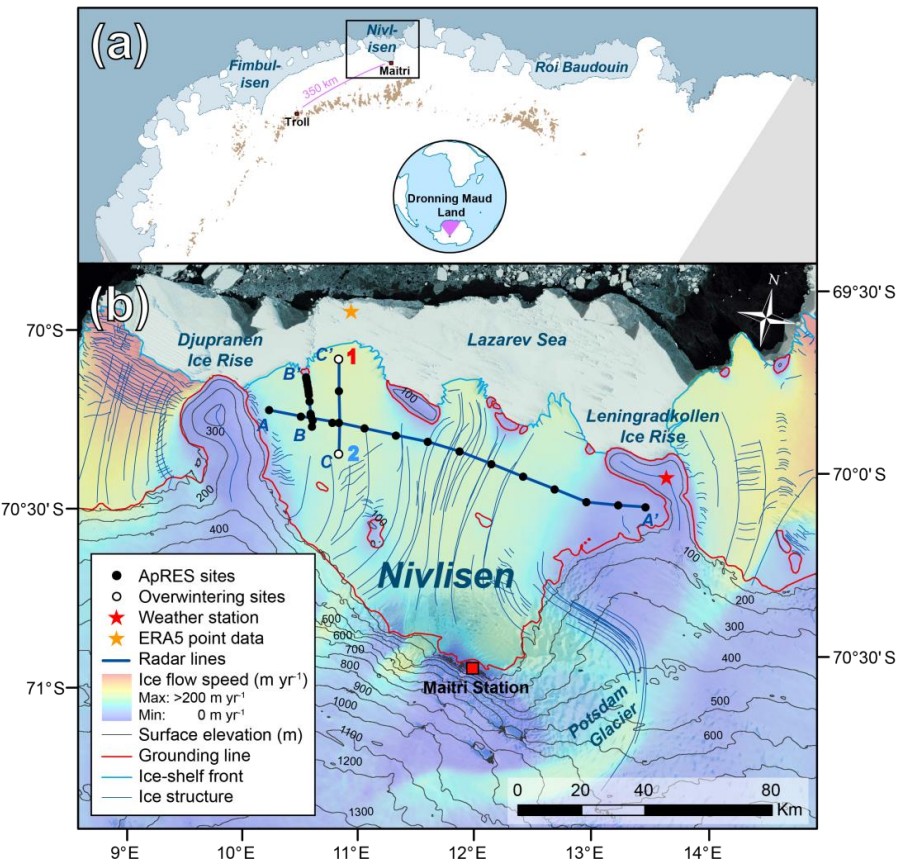

**Figure 1**. Study area: **(a)** Dronning Maud Land coast, with Maitri Station, Troll Station, Nivlisen, Fimbulisen, and Roi Baudouin Ice Shelf. **(b)** Nivlisen ice shelf with surrounding areas. Study sites, where ApRES and stakes for ice speed and surface mass balance were made, ApRES overwintering sites (no. 1 called "seaward" and no. 2 called "landward"), and low-frequency radar profiles (A, B, and C). Satellite derived ice speed (Rignot et al., 2011), surface elevation (m a.s.l.; Howat et al., 2019), grounding line, ice-shelf front (Mouginot et al., 2017), and ice structure (Goel et al, in review) are also shown. Background image is Landsat image mosaic (Bindschadler et al., 2008) and grid coordinate system WGS-84.

In this study, we measured basal melting at Nivlisen (70° S, 12° E), Dronning Maud Land, East Antarctica, using autonomous phase-sensitive radio-echo sounders (ApRES; Fig. 1). Phase-sensitive radars uses a technique where the phase of individual internal ice reflectors is tracked, yielding time series of ice thickness change at high-resolution (~1 mm) over short time



intervals (Corr et al., 2002; Nicholls et al., 2015) and have been used to measure basal and
englacial properties of ice at several locations around Antarctica (e.g., Davis et al., 2018; Jenkins
et al., 2006; Marsh et al., 2016; Stewart et al., 2019), and recently also in Greenland (Vaňková et
al., 2018). Our objective is to study the spatial and temporal variations of basal melting to
explain them using: (1) radar profiles of ice thickness, (2) in situ measured and satellite-derived
ice flow speed and surface mass balance, (3) atmospheric forcing from reanalysis data, sea-ice
distributions, and ocean tides. The data imply that different melt modes were present at Nivlisen.
Our in situ measured data of basal melting complements satellite-derived maps of spatially-
smoothed time-averaged melt rates, and will be a valuable source of data for validation of ice
shelf and ocean models.

## 2  Study area

In the following section, we summarize the geographical, glaciological, and
oceanographic settings of the study area. Dronning Maud Land covers a large area of East
Antarctica, and its 2000-km-long coast is characterized by extensive ice shelves interspersed
with numerous ice rises and ice-sheet promontories (Fig. 1a). Individual ice shelves are relatively
small, but extend close to, or even beyond, the continental-shelf break (Heywood et al., 1998).
Satellite-derived ice-shelf averaged net basal melt rates in Dronning Maud Land vary from near
zero to 7 m yr$^{-1}$ (2003 to 2008; Rignot et al., 2013). The interior of this region is partly separated
by high mountains, causing steep slopes from the continental plateau towards the coastal areas
(Howat et al., 2019). Nivlisen is located in central Dronning Maud Land, 400 km east of
Fimbulisen (Fig. 1a), the largest ice shelf in the area. 100 km south of Nivlisen lies the Wohlthat
Massif, with a maximum elevation of ~3000 m above sea level (a.s.l.). Between the mountain
massif and the ice shelf lies the Schirmacher Oasis, an ice-free area with a maximum elevation of
~250 m a.s.l., with numerous lakes and ponds. The drainage basin of Nivlisen (27 700 km$^2$),
including the grounded ice that drains to the ice shelf, has an ice volume equivalent to 8 cm of
global sea-level rise (Rignot et al., 2019).
Nivlisen has an areal extent of ~7300 km$^2$ and forms a closed embayment between two
larger promontory-type ice rises, Djupranen and Leningradkollen (Fig. 1b). Ice rises are
locations where ice-shelf flow is diverted around the grounded ice and are miniature ice caps



with their own flow fields from the summit (Matsuoka et al., 2015). Ice rumples are smaller
features that impose a disturbance on the ice shelf flow, causing the ice to thicken upstream with
extensive crevassing in the grounding zone. Such grounded features are known to play vital roles
in ice-shelf and ice-sheet dynamics over various timescales. For example, un-grounding of an ice
rumple within the ice shelves of Pine Island and Thwaites Glacier is thought to be a major cause
of the ongoing rapid retreat and thinning (Favier et al., 2012; Gladstone et al., 2012; Jenkins et
al., 2010). Bawden Ice Rise near the edge of the Larsen C Ice Shelf helps maintain the shelf,
despite the collapse of neighbouring Larsen A and B ice shelves (Borstad et al., 2013; Holland et
al., 2015). Nivlisen is grounded at a series of smaller ice rises and rumples near the present ice
front, as well as at a few ice rumples in the middle of the ice shelf (Moholdt and Matsuoka,
2015). The bathymetry under the ice shelf is unknown.

The average ice shelf flow speed is 80 m yr$^{-1}$ (Rignot et al., 2011). Potsdam Glacier

drains into Nivlisen from the southeast, with an average ice thickness of  ~1000 m (Fretwell et
al., 2013) and ice flow speed of ~50 m yr$^{-1}$ (Anschütz et al., 2007; Rignot et al., 2011). The
satellite-derived estimate of the grounding-line flux for Nivlisen was 3.9 ± 0.8 Gt yr$^{-1}$
(2007−2008; Rignot et al., 2013). Elevated topography of the ice rises causes highly-variable
local climate and surface mass balance (Lenaerts et al., 2014). In addition, Nivlisen has large
surface mass balance transitions from being positive in the firn area near the ice front to being
negative in the blue-ice area near the grounding zone, with increased wind erosion, evaporation,
and sublimation due to katabatic winds (Horwath et al., 2006). Near the grounding zone, summer
surface melting is sufficient to form supraglacial lakes and streams that may occasionally drain
through the ice shelf (Kingslake et al., 2015), making Nivlisen potentially sensitive to
hydrofracturing (Lenaerts et al., 2017). Rignot et al. (2013) estimated the surface mass balance to
be 1.8 ± 0.3 Gt yr$^{-1}$ (average 1979−2010) and the average calving flux to be 1.3 ± 0.4 Gt yr$^{-1}$
(2007−2008). Together with the grounding-line flux mentioned earlier and a slightly positive net
mass balance of 0.6 Gt yr$^{-1}$ (2003−2008) results in a residual net basal melt of 3.9 Gt yr$^{-1}$, or an
average basal melt rate of 0.5 ± 0.2 m yr$^{-1}$ (Rignot et al., 2013). Thus, basal melting comprises
~75 % of the total mass losses, with the remaining ~25 % coming from iceberg calving.

The continental shelf extends ~100 km north of Nivlisen into the Lazarev Sea, and is

roughly 500 m deep (Arndt et al., 2013). Carbon dating of laminated sediments near the ice shelf



suggests that the continental shelf was deglaciated ~11 kyr ago (Gingele et al., 1997). At the
eastern border of Lazarev Sea lies Astrid Ridge (~12° E), an undersea bathymetric feature
extending from the Antarctic margin northward to ~65° S. Farther east lies Gunnerus Ridge
(~33° E), where circumpolar deep water is entrained, which is then cooled and modified to
become warm deep water (Dong et al., 2016; Ryan et al., 2016). Warm deep water flows
westward along the continental slope and is entrained into the Weddell Gyre. The Southern
Ocean, including the Weddell Sea, has warmed over recent decades (Gille, 2002; Schmidtko et
al., 2014) with the changes driven primarily by anthropogenic climate warming (Swart et al.,
2018). Sea-ice cover has increased slightly since 1979 around Antarctica in general (De Santis et
al., 2017), however extreme changes have occurred in recent years with record maxima three
years in a row (2012 to 2014), followed by record summertime minimum in 2016 and 2017
(Shepherd et al., 2018; Stuecker et al., 2017; Turner et al., 2015). Sea-ice fluctuations are
strongly correlated with the dominant modes of Southern Hemisphere climate variability (Kwok
et al., 2016; Kwok and Comiso, 2002), although further studies are needed to understand the
drivers behind these fluctuations (Turner 2017). An increase in the seasonality of the easterly
winds has been observed (Hazel and Stewart, 2019) and this may affect the formation and export
of sea ice and the transport of surface waters and warm deep water to the continental shelf. All
these pan-Antarctic observations may affect ocean water flow and then ice-shelf thinning in
Dronning Maud Land, which remain largely unknown.

**3   Data and Methods**

We conducted three field campaigns on Nivlisen and adjacent ice rises during Antarctic

austral summers, from mid-November until end of December, 2016 to 2018, with logistic
support from the Indian Maitri Station and Norwegian Troll Station (Fig. 1a). We measured basal
melting under Nivlisen using two ApRES systems (200–400 MHz), developed by the British
Antarctic Survey (British Antarctic Survey, 2018; Nicholls et al., 2015). Below, we describe the
methods used to collect the ApRES data and process them to derive basal melt rates (Sect. 3.1).
We also studied the ice-shelf thickness and basal structure with a low-frequency (5 MHz) radio-
echo sounder (Sect. 3.2). Finally, the annual ice flow speed and surface mass balance were
measured at stake locations along three profiles on the ice shelf (Sect. 3.3).



## 3.1    Autonomous phase-sensitive radar

In December 2016, we installed stakes for measurement of ice speed and surface mass balance at 29 locations on Nivlisen and measured the ice thickness using an ApRES system (Fig. 1b): (A) 13 stakes were placed across the ice shelf at a spacing of 10 km (profile A), (2) 10 stakes were placed along the ice flow towards a grounded feature near the ice front with a spacing of 1 to 4 km (profile B), and (3) Four stakes were placed along the ice flow out on an ice tongue at a spacing of 10 km (profile C). After the initial measurements, we installed similar ApRES systems at two locations for hourly measurements of basal melting and strain rates over the winter, each powered by a 12 V battery (Fig. 1b): (1) 4 km from the ice-shelf front, called the "seaward site" hereafter, and (2) 35 km from the ice shelf front, called the "landward site". In December 2017 and 2018, we revisited and re-measured all stake sites to get annual averaged values of basal melting and strain rates and retrieved the time-series data from the two overwintering stations. Extensive crevassing prevented the three sites closest to the ice rumple (profile B, Fig. 1b) from being revisited in 2018.

ApRES uses the frequency-modulated continuous wave (FMCW) technique (Rahman, 2016). The instrument transmits a signal sweeping from 200–400 MHz over a period of 1 s to form a chirp (Nicholls et al., 2015). The system has a low-power consumption, with a power to the transmitter antenna of 100 mW. The averaged signal was amplified and de-ramped, a process where the received signal is mixed with a replica of the transmitted signal to extract differences in frequencies. The de-ramped signal was then filtered to amplify the higher frequencies preferentially, which enhanced weaker signals from more distant reflectors. Each sample consisted of 100 chirps, collected over a period of a few minutes. The data were digitized and stored on secure digital cards for further processing.

We processed the data following Brennan et al., (2014) and Nicholls et al. (2015) (Supplements Fig. S1). The data were Fourier transformed to give a complex signal amplitude as a function of delay time, or depth, assuming a constant propagation velocity of 168 m $\mu s^{-1}$. An amplitude cross correlation between the two returns for a depth range within the firn layer (typically from 40 to 70 m) provided a vertical shift that that approximately accounted for snow accumulation between the visits. The displacement of the reflectors between the two visits were then plotted as a function of depth. To give the necessary depth resolution, the phase of the



signals was used to calculate the displacements by cross-correlating 4 m segments of the first
profile with the complex conjugate of the corresponding segment of the second. Under the
assumption of a constant vertical strain rate between the bottom of the firn layer and just above
the ice base, we fit a straight line to the layer displacements. The effect of the correction for
snow accumulation between the two visits, both the coarse correction mentioned above and the
precise correction inherent in the phase processing, and the effect of the non-linear (with depth)
displacements due to firn compaction, are both contained within the intercept at the vertical axis.
Thus the basal melt is given by the deviation of the displacement of the basal reflection from the
straight line fit (Supplements Fig. S1). The error in the calculated strain was estimated using the
quality of fit of the linear regression. The uncertainty in the melt rate was obtained by combining
the uncertainty in the strain rate with the uncertainty in the change in the range to the basal
reflector, deduced from the signal-to-noise ratios of the two basal reflections.
To calculate the hourly time-series data from the two overwintering sites (Fig. 1b), we
tracked the basal returns using phase-coherent processing, allowing us to determine the speed of
motion of the ice base with respect to the antenna, which hereafter is called the thinning rate. To
remove the component of ice-column vertical strain rate caused by tidal variations, we filtered
the basal vertical speeds with a 36 h low-pass filter. We removed an annual average vertical
strain rate from the filtered basal motion, resulting in net melt rates. We assumed that at periods
longer than 36 hours, the variability in strain rate is small compared with the basal melt rate and
varies on a much longer timescale than those of interest here.
3.2  Low-frequency radar profiling
In December 2016, we collected ~180 km of continuous radio-echo sounding profiles
across the ice flow (profile A) and along the ice flow (profile B and C) of Nivlisen to measure
ice thickness and englacial and basal structure (Fig. 1b). We used a common-offset impulse radar
system (Dowdeswell and Evans, 2004) based on the radar developed by Matsuoka et al. (2012)
and processing steps following Lindbäck et al. (2014). We used half-wavelength dipole antennas
with a 5 MHz centre frequency, with a Kentech impulse transmitter with an average output
power of 35 W. The transmitter and receiver systems were mounted on two sleds and towed
behind a snowmobile at a speed of $\sim$10 km h$^{-1}$. We positioned the traces using data from a code-
phase global positioning system (GPS) receiver mounted on the radar receiver box 20 m in front



of the common mid-point of the antennas along the travelled trajectory of the snowmobile. We
post-corrected the height using the Canadian precise point-processing service (CSRS-PPP;
Natural Resources Canada, 2017) from a kinematic carrier-phase dual-frequency GPS receiver
mounted on the snowmobile. The radar measurements had an average line spacing of ~5 m.

Several corrections and filters were applied to the radar data: (1) dewow and bandpass

filters, to remove unwanted frequency components in the data, (2) depth-variable gain function,
and (3) normal move-out correction to correct for antenna separation, including adjusted travel
times for the trigger delay. The basal returns were digitized semi-automatically with a cross-
correlation picker at the first break of the bed reflection (Irving et al., 2007). We calculated ice
thickness from the picked travel times of the bed return using a constant radio-wave velocity of
168 m $\mu$s$^{-1}$ for ice. We added a correction term of 2 m to account for the faster propagation in
the firn based on the snow density (Sect. 3.3). The firn has a depth of ~50 m, derived from the
ApRES internal reflectors. Ice draft was calculated from the ice thickness by subtracting the
surface elevation, corrected for local sea level (freeboard). We estimated the error in ice
thickness by standard analytical error propagation methods (Lapazaran et al., 2016; Taylor,
1996), outlined in Lindbäck et al. (2018). The estimation included the error in the radar
acquisition and horizontal positioning error, where the radar acquisition errors comprised errors
in radio-wave velocity and two-way travel time. Velocity can vary spatially, depending mainly
on density. Errors in two-way travel time were estimated to be the range resolution, which is the
accuracy of the measurement of the distance between the antenna and the bed. The average radar
system error was estimated to $13.3 \pm 1.2$ m. The surface and base of the ice shelf is relatively
flat, giving very small errors in horizontal positioning ($0.1 \pm 0.2$ m). The total error in ice
thickness is presented together with the data in Sect. 4.
3.3    Ice flow and surface mass balance from stakes

We measured ice flow and surface mass balance at all 29 ApRES stakes on Nivlisen.

Stake height over the surface was measured manually, and stake position was measured statically
for 15 minutes using carrier-phase dual-frequency GPS receivers at 1 s logging interval. The
stakes were revisited and measured in December 2017 and 2018. We processed the positions
using CSRS-PPP. Snow density was measured at five locations on Nivlisen with an auger drill to
a depth of 3 m and varied from 430–450 kg m$^{-3}$. We used the average snow density of 440 kg





m$^{-3}$ and an ice density of 917 kg m$^{-3}$ to calculate the surface mass balance in ice equivalent. Ice
flow speed and surface mass balance were compared with estimates from satellite data (Rignot et
al., 2011) and regional atmospheric modelling (van de Berg et al., 2006).

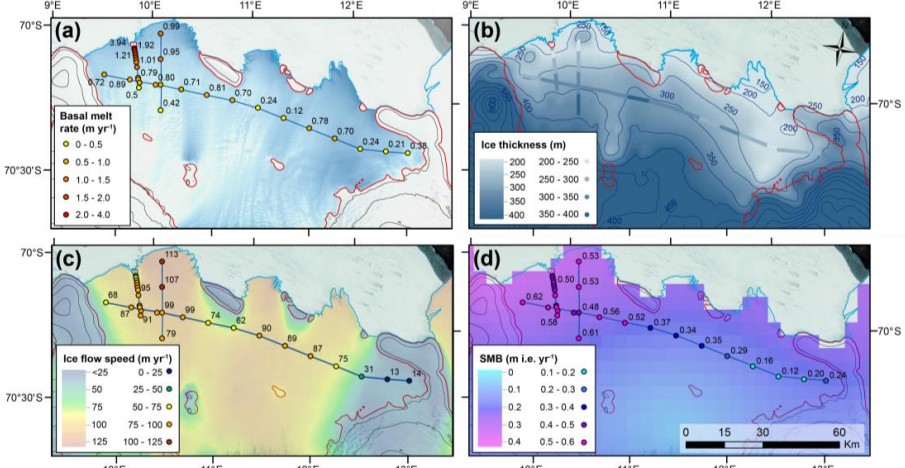


**Figure 2.** Stake and profile measurements: **(a)** ApRES-derived annual averaged basal melt rates for 2017 and hill
shade (blue) extracted from the Reference Elevation Model of Antarctica (REMA; Howat et al., 2019). See
Supplements Fig. S2 for annual averaged basal melt rates for 2018, which is on average within ±10 % from the 2017
values. **(b)** Ice thickness from low-frequency radar profiles (point values) and from Bedmap2 product (grid and
contour lines; Fretwell et al., 2013). **(c)** Ice flow speed from stakes and gridded satellite values (Rignot et al., 2011).
**(d)** Surface mass balance (SMB) from stakes and gridded modelled values (Le Brocq et al., 2010). Contour lines and
background images are the same as in Fig. 1.

**4    Results**

In 2017, annual averaged melt rates at 29 stake locations on Nivlisen (Fig. 1b) ranged

from 0.12 ± 0.06 to 3.94 ± 0.04 m yr$^{-1}$ (Fig. 2a and Fig. 3), with a median value of 0.80 m yr$^{-1}$.
The highest annual averaged melt rates were observed close to an ice rumple at the ice front and
the lowest melt rates in the central and eastern parts of the ice shelf. In 2018, annual averaged
melt rates at 26 locations, ranged from 0.13 ± 0.06 to 1.48 ± 0.01 m yr$^{-1}$, excluding three sites
closest to the ice rumple, with high melt rates in 2017 (Supplements Fig. S2). The median melt
rate in 2018 was 0.72 m yr$^{-1}$. Melt rates were slightly lower in the second year at 18 sites and for



8 sites slightly higher. Excluding the three sites closest to the ice rumple in 2017, annual basal
melt rates between 2017 and 2018 differ by ± 0.1 m yr$^{-1}$. Errors in melt rates were on average
0.023 m yr$^{-1}$ in 2017 and 0.025 m yr$^{-1}$ in 2018.

Strain rates were in general low, having a median annual averaged value of −4.7 x 10$^{-4}$

yr$^{-1}$ in 2017 and −4.6 x 10$^{-4}$ yr$^{-1}$ in 2018. The vertical strain-rate contribution to the average rate
of change was on average 22 %. The errors in strain were low, on average 6.2 x 10$^{-5}$ yr$^{-1}$ in 2017
and 7.1 x 10$^{-5}$ yr$^{-1}$ in 2018. For most parts of the ice shelf the strain rates were negative, meaning
that the ice was thinning by longitudinal stretching, however, close to the ice rumple mentioned
earlier (profile B; Fig. 3) we observed a transition from negative to positive strain rates (from
−5.4 x 10$^{-4}$ to 2.2 x 10$^{-2}$ yr$^{-1}$), with increasing compressional thickening of the ice towards the
ice rumple. Positive strain rates were also observed for five sites 5−10 km upstream of the larger
ice rises in the central and in the eastern part of the ice shelf (profile A; Fig. 3), indicating a far-
reaching buttressing effect (distance up to ∼30 ice thicknesses).

The two overwintering ApRES systems were used to derive time series of melt rates. The

seaward overwintering site was located 4 km from the ice front and had an ice draft of 130 m,
measured with low-frequency radar. It operated for 14 months (from 11 Dec 2016–4 Feb 2017)
before the battery failed. Thirty-six hour low-pass filtered melt rates at this site varied from ∼0 to
5.6 m yr$^{-1}$, where the highest melt rates occurred in summer (29 Jan 2017; Fig. 4a). The
landward overwintering site was located 35 km from the ice front and had an ice draft of 280 m.
The data cover 23 months (from 4 Jan 2017–27 Nov 2018), except for December 2017 when the
instrument was used for measuring annual melt rates at other locations. At this site, 36 h low-
pass filtered melt rates varied from ∼0 to 2.0 m yr$^{-1}$, where the highest melt rates occurred  in
winter (12 Jun 2018; Fig. 5a).

Ice thickness, measured with low-frequency radar along profiles A, B, and C (Fig. 1b),

varied from 160 to 330 m (Fig. 2b), with a median value of 260 m. We observed the thinnest ice
close to the ice front along profile C (Fig. 3) and the thickest ice in the southern-most part of the
ice shelf along the same profile. The total error in ice thickness along the profiles, including
radar system and positioning errors, varied between 10.6 and 15.7 m. The broad thickness pattern
agreed with satellite-derived freeboard estimates from Bedmap2 (Fretwell et al., 2013), except



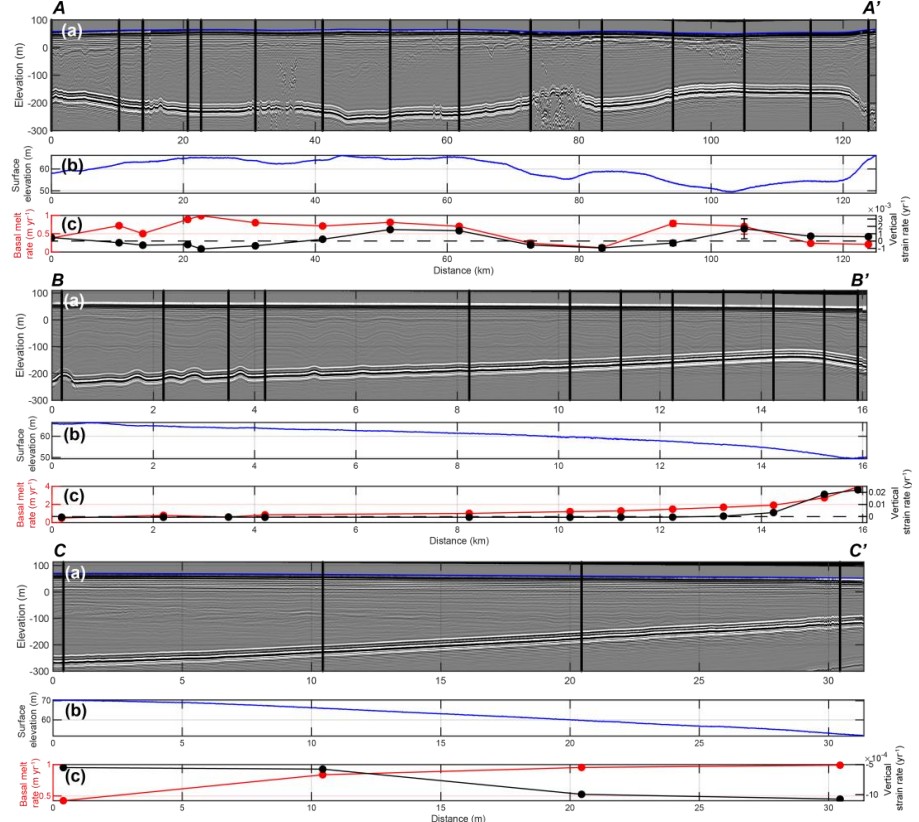

**Figure 3**. Profiles of low-frequency radar, ice surface elevation, basal melt, and strain (locations in Fig. 1b): A–A'

across ice flow from west to east (125 km), B–B' along ice flow from south to north towards an ice rumple (16 km),

and C–C' along ice flow from south to north out on an ice tongue (32 km). Sub-panels show **(a)** radar profiles with

surface elevation (blue line), englacial stratigraphy, and basal elevation (grey tone shading), and locations of ApRES

measurements (black vertical lines), **(b)** surface elevation from carrier-phase kinematic GPS measurements, and **(c)**

annual basal melt rate (red) and vertical strain rates (black, dashed = 0) for 2017. Note that the x-axis scales vary

between the three profiles. Surface elevation is referenced to local sea level (freeboard).

on the western ice tongue (profile C), where the thickness reported in Bedmap2 is too high (Fig.

2b). Ice draft varied from 120 to 280 m with a median value of 220 m (Fig. 3). We observed no

significant relation between basal melting and ice draft. Several locations with undulating

englacial layers, basal channels and crevasses were visible in the radar profiles (Fig. 3). Stake-



measured ice flow speeds varied from 13 to 113 m yr$^{-1}$ in 2017, with an average value of 80 m
yr$^{-1}$, agreeing with satellite estimates (Rignot et al., 2011; Fig. 2c). Surface mass balance values
varied between 0.12 and 0.62 m i.e. yr$^{-1}$ in 2017 with an average of 0.45 i.e. yr$^{-1}$, higher than the
modelled average estimates of 0.2 m i.e. yr$^{-1}$ (van de Berg et al., 2006), but with the same spatial
pattern (Fig. 2d).

**5    Discussion**
In the following sections, we discuss the spatial (Sect. 5.1) and temporal (Sect. 5.2)
variations in basal melting and compare our results with other studies from Antarctica. For each
section, we also discuss strengths, limitations, and recommendations for future studies.
5.1    Spatial variations in melting
On Nivlisen, we observed the highest annual averaged melt rates (3.9 m yr$^{-1}$) close to a
small (4.2 km$^2$) ice rumple at the ice front (Fig. 2a and Fig. 3). Similar high melt rates (~4 m
yr$^{-1}$) were inferred from satellite data nearby Bawden Ice Rise (Adusumilli et al., 2018). In
modelling experiments, the higher melt rates arose from the generation of energetic short-length-
scale diurnal topographic vorticity waves (Mueller et al., 2012). This required a thin water
column (shallow bathymetry) under the ice-shelf. At Nivlisen, we have no observations of tidal
strengths near the ice rumple, but the bathymetry must be shallow since the ice shelf grounds in
this region. Ice shelf thinning could potentially increase the water column depth and have a
negative (stabilizing) feedback on the melting, reducing the topographic vorticity waves (Mueller
et al., 2012, 2018; Padman et al., 2018), though no clear relationship was found between ice draft
and basal melting rates in our study. In terms of ice thickness change, the observed thinning from
the basal melt is compensated by a positive vertical strain that implies compressional thickening
towards the ice rumple (up to 4 m yr$^{-1}$). Thicker ice towards the ice rumple indicates a
buttressing effect on the ice shelf (profile B; Fig. 3). We observed many crevasses in this region
that made it, for safety reasons, difficult to revisit the three closest sites during the third field
season (Dec 2018). The effects of sustained high melt rates at the Nivlisen ice rumple are
uncertain, and modelling work may indicate whether un-grounding of the ice would potentially
lead to substantial loss of buttressing (Borstad et al., 2013).



Estimates of basal melt rates for Dronning Maud Land ice shelves have mainly used
satellite techniques, modelling, or limited spatial or temporal coverage of in situ radar
observations (Berger et al., 2017; Langley et al., 2014b). Fimbulisen is situated 400 km west of
Nivlisen (Fig. 1a) at the outlet of Jutulstraumen, one of the largest ice streams in Dronning Maud
Land. Below the deep keel from Jutulstraumen (300–400 m ice draft), time-averaged melt rates
of several meters per year were observed, whereas at the shallower parts of the ice shelf
(200–300 m ice draft), lower melt rates were observed (Langley et al., 2014a). In addition,
annual-average melt rates were modelled to be near zero for large areas (Hattermann et al.,
2014). Hattermann et al. (2014) hypothesized that melt mode 1 occurred at the deepest parts of
Fimbulisen (below ice drafts of 400 m). The rising melt plume caused marine accretion at
shallower depths closer to the ice front, together with seasonal mode-3 melting, resulting in the
low net melt rates, with seasonal marine ice formation being inferred from an ice shelf cavity
mooring (Hattermann et al., 2012). Nivlisen is in comparison relatively thin (Fig. 2b) and we
have no melt observations from the thicker ice in the southern areas. Grounding line ice drafts
(Fig. 1b), derived from Fretwell et al. (2013) and Mouginot et al. (2017), have an average value
of 350 m. The deepest part of the grounding line (630 ± 100 m) is located at the outflow of
Potsdam Glacier (Fig 1b), where higher melt rates may occur. In addition, Nivlisen has three ice
tongues, separated by ice rises and ice rumples, where the ocean can gain access to the inner
parts of the ice shelf cavity. At Fimbulisen, Hattermann et al. (2012, 2014) found that a portion
of the westward flowing coastal current was diverted under the ice shelf between two ice rises.
Similar inflow pathways may also exist beneath the ice tongues of Nivlisen, explaining the
variations of melt rates along profile A (Fig. 2a). At Fimbulisen, high melt rates (3 m yr$^{-1}$) were
also observed and modelled close to the ice front at shallow depths (< 200 m; Hattermann et al.,
2014; Langley et al., 2014b), which is consistent with our results.
In the low-frequency radar profiles, we observed several undulating ice-base features
(profile A and B; Fig. 3), where the englacial layers warp downwards, which is likely an
indication of basal channels or crevasses. The southernmost measurement in profile B is located
at one of these down-warping features, where surface elevation is slightly lowered locally (−0.5
m). Higher melt rates were not observed here compared with the surrounding sites, although,
higher melt rates typically occur on the flanks of basal channels, rather than at their apex (Berger



et al., 2017). The channel may have formed at an upstream ice rumple and been passively
advected downstream (Fig. 2a). Basal channels are important features influencing the ice-shelf
stability, since they affect ice-shelf cavity circulation and play a role in the exchange of heat and
mass between the ocean and ice shelf (Gladish et al., 2012; McGrath et al., 2012; Millgate et al.,
2013; Stanton et al., 2013). Basal channels are not restricted to rapidly melting ice shelves and
have been observed elsewhere in Dronning Maud Land, at Fimbulisen (Langley et al., 2014a)
and Roi Baudouin Ice Shelf (Fig. 1a; Berger et al., 2017). Detailed studies of these features
together with basal melting are needed to understand their initiation, evolution, and role in the
overall mass balance of ice shelves (Alley et al., 2016).

## 5.2 Temporal variations in melting

Melt rates at Nivlisen varied on a broad range of timescales (Fig. 4 and 5). At the
seaward site, we observed a seasonal signal, where the monthly averaged melt rates were two to
three times higher in the summer than in winter (Fig. 4a, Supplements Fig. S3). At the landward
site, we observed no seasonal pattern, however, some variability on monthly time-scales was
present (Fig. 5a, Supplements Fig. S3). We performed a continuous wavelet transform on the
time-series data from the two overwintering sites, based on the method and software package
provided by Grinsted et al. (2004). The wavelet transform is used to study localized intermittent
periodicities, in contrast to more traditional mathematical methods, such as Fourier analysis,
which assumes that the underlying process is stationary in time. We used a Morlet wavelet with
$\omega_0 = 6$, which provides a good balance between time and frequency localization. The wavelet
transform shows the normalized thinning rates at different scales to identify dominant periods of
variability in time (Fig. 4b, 5b). The statistical significance was assessed relative to the null
hypothesis, modelled by a first order autoregressive process. The wavelet transform has edge
artefacts since it is not completely localized in time, as indicated by the cone of influence,
masking out low frequency signals at the beginning and end of the time series.  The thinning
variability at diurnal timescales, and to some extent semi-diurnal timescales, varied at an
approximately two-weekly period. This reflects the fortnightly spring-neap tidal cycle at which
the strength of the tidal currents varies due to the interference of different constituents, usually
$M_2$ and $S_2$ in this area (plotted as white dashed lines in Fig. 4b and 5b). Stronger tidal currents
increase the heat exchange at the ice-ocean interface and may hence cause more rapid melt. At





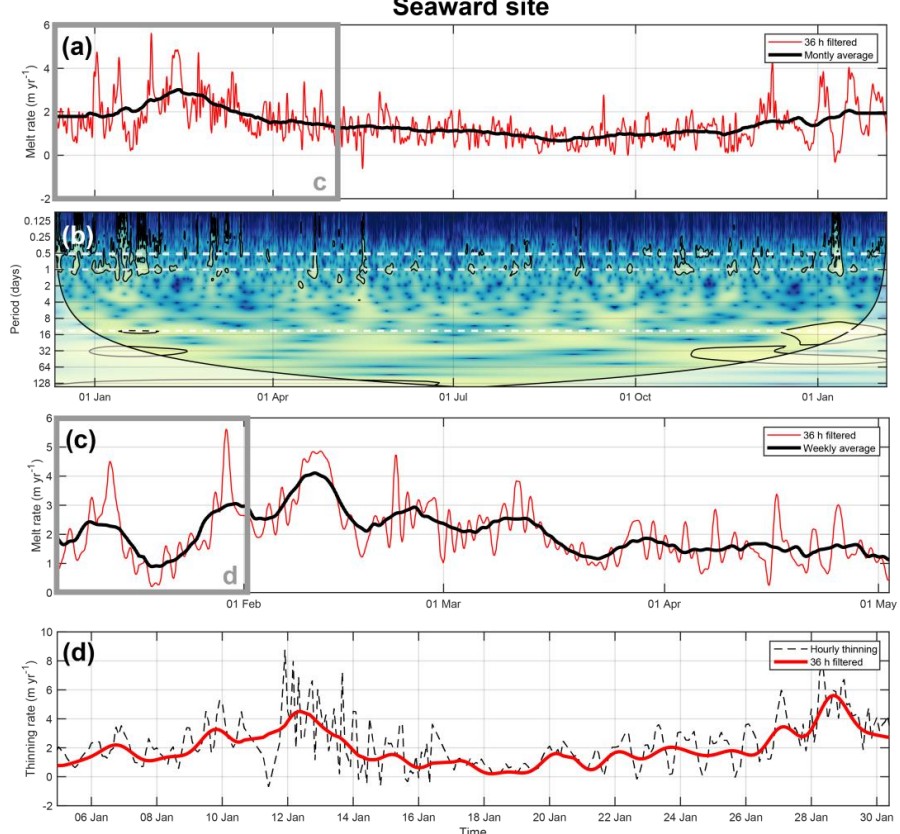


**Figure 4.** Basal melt and thinning rates for the seaward overwintering site, with variations on time scales of **(a)** months (11 Dec 2016−4 Feb 2017), **(c)** weeks (1 Jan−1 May 2017), and **(d)** days (1−31 Jan 2017). Dashed black line in (d) is the unfiltered raw data with thickness change including strain rates. **(b)** Continuous wavelet transform of the normalized thinning to identify the dominant modes of variability at different time scales. The left axis is the Fourier period. The colour shading represents the thinning associated with fluctuations over the course of the year with a particular time period (yellow = high power, blue = low power). The black contours delimit significant modes of variance at 95 % against red noise. Within the cone of influence, shown as a lighter shade, edge effects become important. Dashed white lines show the periods of major tidal constituents (0.5 d ≈ $K_1$, 1 d ≈ $M_2/S_2$, and 14 d ≈ $M_f$).


periods shorter than 36 hours, however, we cannot differentiate the strain signal from the melt
signal. We also see some evidence of a slower variability in data centred on 2−4 days (Fig. 4d
and 5d), which may be a result of mesoscale activity passing by the site (eddies or internal
waves), which then show up in the melt rate. This is to some extent supported by Fourier

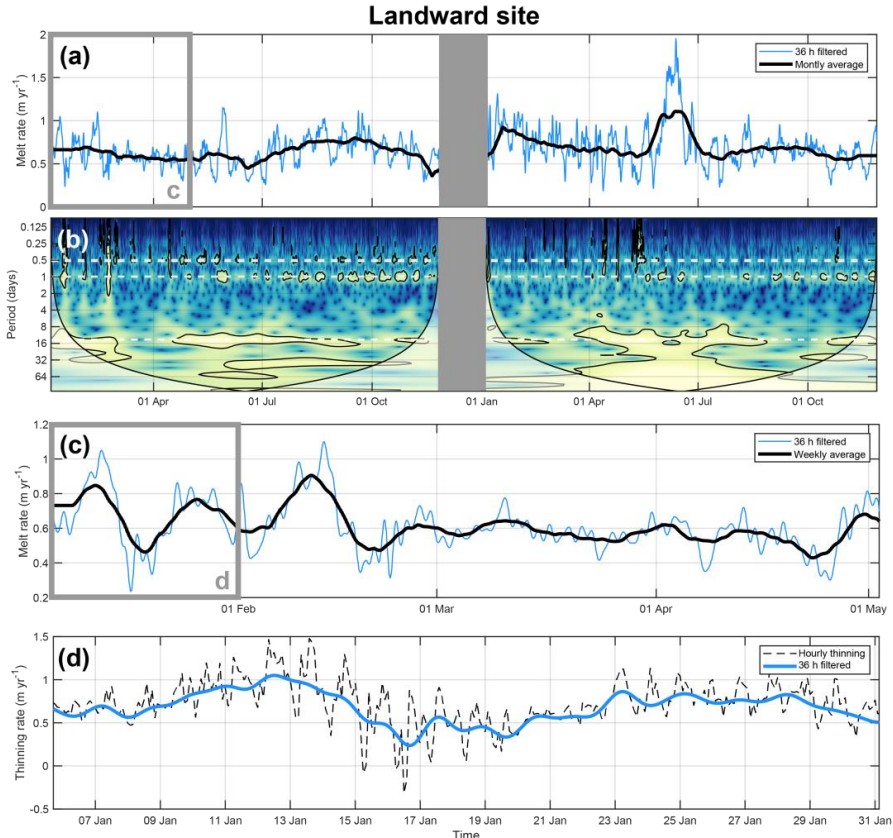

**Figure 5.** Basal melt and thinning rates for the landward overwintering site, with variations on time scales of **(a)** months (4 Jan 2017–27 Nov 2018), **(c)** weeks (4 Jan–1 May 2017), and **(d)** days (4–31 Jan 2017). **(b)** Continuous wavelet transform as described in Fig. 4. Grey box masks a time period with no data. All legends are the same as Fig. 4.

analysis of the normalized 36 h filtered melt rates, which show peaks in power spectral density at 2–4 days, mostly visible at the seaward site (Supplements Fig. S4).

At the landward site, we observed no increased melting in summer, but we observed one melt peak in winter (12 June 2018; Fig. 5a). The melt event may have been caused by pulses of modified warm deep water reaching the base of the ice shelf as described by Hattermann et al. (2012), but it could also relate to other mesoscale activities within the cavity. In any case, the isolated event and the generally low melt rates suggest that warm deep water had limited access



to the base of Nivlisen during 2017 and 2018. The observation is consistent with earlier studies,
showing that ice shelf cavities in this region are mainly filled with cold and fresh eastern shelf
water (Nicholls et al., 2006; Thompson et al., 2018). Along most of the Dronning Maud Land
coast, the ice shelf cavities are separated from warm deep water by the Antarctic slope front,
which is a pronounced transition zone over the narrow continental shelf between eastern shelf
water and warm deep water, mainly attributed to coastal downwelling caused by the prevailing
easterly winds (Sverdrup, 1954; Thompson et al., 2018). Many factors control the extent to
which warm deep water can access the ice shelf cavities, such as the stability of the Antarctic
slope front, local circulation, and bathymetry. The coastal dynamics that set the warm deep water
depth along the continental shelf break involves the balance between wind-driven Ekman
overturning and counteracting eddy fluxes (Nøst et al., 2011; Thompson et al., 2014). These
processes respond to changes in wind and buoyancy fluxes (Hattermann et al., 2014; Stewart and
Thompson, 2016), including self-amplifying feedback of increased fresh water input from
increased basal melting (Hattermann, 2018).

We studied the coherency between the two overwintering melt sites in a wavelet

coherence (Grinsted et al., 2004) for the overlapping time periods in 2017 (Fig. 6). The wavelet
coherence analysis finds significant coherence even if the common power is low, and it shows
significant confidence levels against red noise backgrounds. Locally phase-locked behaviour can
also be revealed; at weekly to monthly periods (7 to 30 days) in summer to fall (Jan−Apr 2017)
the melt rates were in phase, whereas in winter (Apr−Jun) the melting at the seaward site led the
increased signal, preceding the melt at the landward site. In late winter (Sept), the phase shifted
to the landward site leading the melt. At Fimbulisen, the inflow of summer-warmed Antarctic
surface water was observed at moorings close to the ice shelf front with a clear seasonal signal in
water temperatures and salinity (Hattermann et al., 2012). Hattermann et al. (2014) suggested
that Antarctic surface water can reside for several months in the ice shelf cavity, after initially
being subducted beneath the ice front, potentially affecting basal melting deep inside the cavity.
The observed melt rate pattern beneath Nivlisen may be an indication of similar movement of
water masses below the ice shelf and further modelling is needed to study these processes,
currently being hampered by the lack of knowledge of bottom topography beneath the ice shelf.




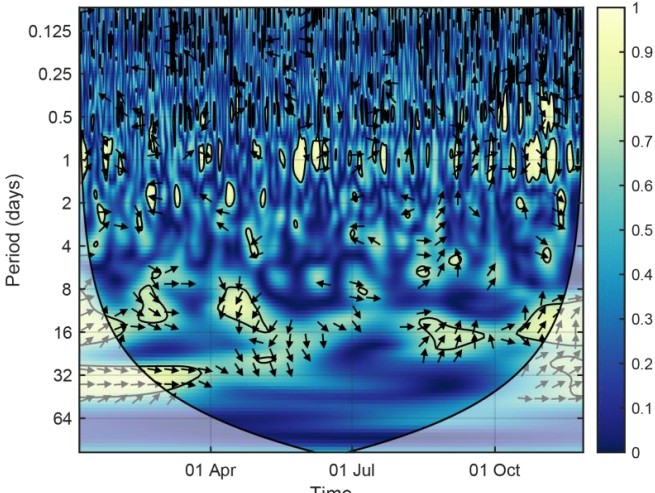


**Figure 6.** Wavelet coherence between the seaward and landward site (4 Jan–27 Nov 2017), showing times where the melt rates have common power. The phase relationship is shown as arrows. At longer periods (8-30 days) in summer to fall (Jan–Apr) the signals are in phase (arrows pointing right), whereas in winter (Apr–Jun) the melt at the seaward site leads the signal (arrows pointing down). In late winter (Sept) the phase shifts to the landward site leading the signal (arrows pointing up).

We compared the basal melt rates with atmospheric ERA5 reanalysis data of wind speed, wind direction, sea-ice cover, air pressure, and temperature (Fig. 7) produced by the European Centre for Medium-Range Weather Forecasts (Copernicus Climate Change Service (C3S), 2017) at a grid point 10 km north of the ice shelf front (Fig. 1b). ERA5 wind speeds at Nivlisen varied on daily timescales, ranging from 0 to 28 m s$^{-1}$. Winds generally blew from the east (Fig. 7b), corresponding to the pressure gradients imposed by the cyclonic system that dominates the Weddell Sea. Wind forcing can play an important role in downwelling and transportation of summer-warmed Antarctic surface water into the ice-shelf cavity (Zhou et al., 2014). We calculated the coherence between the normalized melt rates at the seaward site and wind speeds. The statistical significance level was estimated using Monte Carlo simulation with a Fourier transform method, where a large set of surrogate data set pairs were generated using phase randomization (Schreiber and Schmitz, 2000). In summer, we find a significant coherence between melt rates and wind speeds (r = 0.36, p < 0.05; Supplements Fig. S5). Inspecting individual melt peaks in the summer (dashed vertical lines in Fig. 7) show that they coincide



with higher wind events, and have a time lag of ~0 to 3 days. We found no such coherence in
winter. The variability in winter may be due to the transport mainly dominated by eddies, shed
by instabilities in the along-slope current. Sea-ice cover according to ERA5 decreased or was
absent (defined as less than 15 %) during summer and fall (January to March; Fig. 7c), also
coinciding with higher melt rates. When melt increased in early summer at the seaward site (Dec
2016 and 2017; Fig 4a), we observed less sea-ice cover close to the ice shelf front, which is the
time when solar radiation may warm the surface waters.  Satellite images also show the
variability in sea ice (Supplements Fig. S6), with open water east of the ice tongue in December
2017 that is not resolved in detail in the ERA5 sea-ice cover. The ERA5 air temperatures at 2 m
varied mostly on seasonal time scales, with temperatures between 0 and −10° C in summer, and
down to −24° C in winter (Fig. 7d). The temperature variability in the reanalysis data on shorter
timescales agreed with our weather station on Leningradkollen ice rise (190 m a.s.l.), however,
the seasonal temperature signal had a lower amplitude than at the weather station, which
measured temperatures down to −38° C.

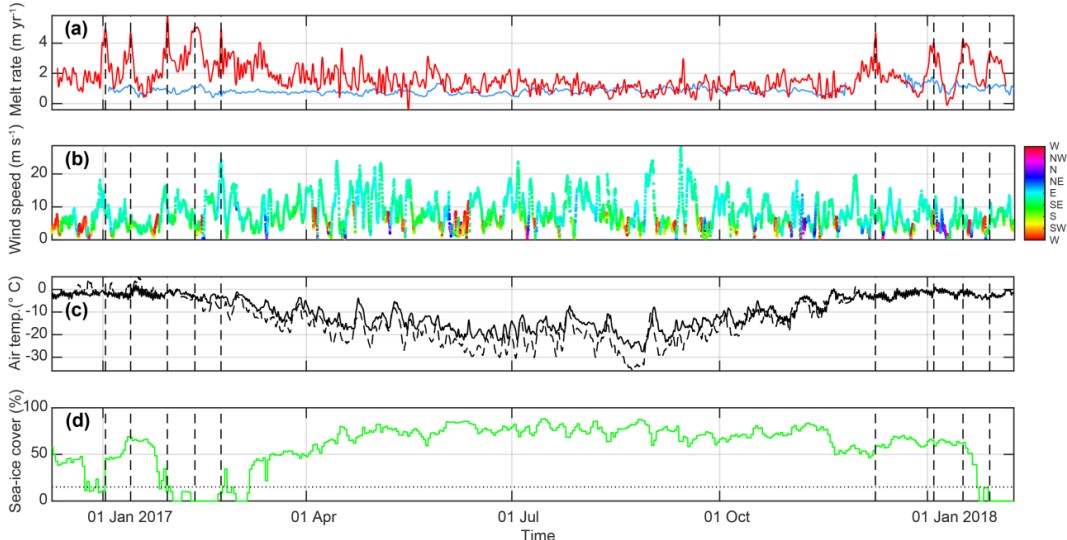


**Figure 7.** Melt peaks compared with atmospheric forcing and sea ice: **(a)** Thirty-six hour low-pass filtered basal
melt rates at seaward site (red) and landward site (blue). ERA5 reanalysis surface data of **(b)** wind speed and
direction, **(c)** 2 m air temperature, where dashed black line is data from a weather station (Fig. 1b), and **(d)** sea-ice
cover, where the dotted line is 15 % (definition of no sea ice). Periods with pronounced basal melting are indicated
with vertical dashed lines for easier comparison.



In summary, the observed higher melt rates during summer and fall correlate with higher
wind speeds, when there is less sea-ice cover and higher air temperatures. We hypothesize that
summer-warmed Antarctic surface water was pushed by wind under the ice shelf at the front.
Warming of the surface water is projected to increase ice-shelf melting along Dronning Maud
Land in future climate scenarios (Kusahara and Hasumi, 2013) and recent studies suggest that
non-linear feedbacks may facilitate an irreversible transition into a state of high melting in the
Weddell Sea (Hattermann, 2018; Hellmer et al., 2017). Surface winds are projected to intensify
over the next century with increased greenhouse gas emissions (Greene et al., 2017) and extreme
changes in sea-ice extent have occurred in recent years (Shepherd et al., 2018). Natural
variability in the atmosphere and oceans remains poorly understood (Turner et al., 2016) and the
declining extent of ice shelves around Antarctica has been ascribed to a complex set of processes
linking the atmosphere, ocean, and sea ice (Adusumilli et al., 2018; Greene et al., 2017).


**6     Conclusions**
We present a two year record of basal melting at Nivlisen, in Dronning Maud Land, East
Antarctica, at high spatial and temporal resolution using in situ phase-sensitive radar
measurements. Annual averaged melt rates are in general moderate, but high melt rates were
observed close to a grounded feature near the ice shelf front. Daily measurements also reveal a
seasonal melt pattern close to the ice shelf front, where the highest melt rates occurred in
summer. Comparing the seasonality in basal melting with forcing from atmospheric reanalysis
data, we found that the variability in the basal melt is likely caused by summer-warmed surface
water pushed by the wind into the ice-shelf cavity. Farther into the ice-shelf cavity, we observe a
different melt regime, with significantly lower melt rates and a clearer tidal signal. We conclude
that warm deep ocean water has a limited effect on the basal melting of Nivlisen, likely because
the present configuration of the Antarctic slope front, which separates the deeper water from the
continent, protects the ice shelf from those warmer water masses.
Our study highlights that, although many of the ice shelves of East Antarctica have
generally low melt rates, their seaward portions remain susceptible to higher rates of melting due
to the influence of summer-warmed surface waters. Reduced sea-ice cover and higher wind





speeds may increase leading mode-3 melting, while weaker winds and/or changes in the surface
buoyancy forcing may increase exposure of the sub ice-shelf cavities to warm deep water and
therefore increase mode-2 melting. Increases in basal melting will tend to thin the ice shelves and
reduce the buttressing on the inland ice sheet. Many ice shelves like Nivlisen are stabilized by
pinning points at their ice fronts, which may be sensitive areas for future change. It remains to be
understood to what extent, increased summer-warmth driven melting, intensified in the vicinity
of these pinning points may affect the ice flow dynamics and ice-shelf stability. Our study shows
the use of and need for continuous in situ monitoring of Antarctic ice shelves to resolve
variability in basal melting that is not captured in satellite data. Long-term, high-resolution time-
series data are important to understand the complex mechanisms involved in ice shelf–ocean
interactions, which in turn is important for ice sheet models.

**Data availability**

The compiled data sets of basal melt, strain rates, ice speeds, surface mass balance, and low-
frequency radar profiles will be available at the Norwegian Polar Data Centre
(https://data.npolar.no).

**Author contribution**

KL led the overall data analysis and interpretations, and prepared the paper with contributions
from all co-authors. KL, GM, and BP collected the ApRES, ice speed and surface mass balance
in the field. KWN was responsible for the ApRES system setup. KM was responsible for the
low-frequency radar system and collected the data in field. TH contributed to the discussion
section. MT and KM were the project leaders.

**Competing interests**

KM is a member of the editorial board of the journal.





## Acknowledgments

This work was part of the MADICE (Mass balance, dynamics, and climate of the central Dronning Maud Land coast, East Antarctica) project, funded by the Research Council of Norway (project 248780) and the Ministry of Earth Sciences, India (project MoES/Indo-Nor/PS-3/2015). Logistic support was provided from Indian Maitri Station and Norwegian Troll Station, Antarctica; we would like to thank the NCPOR and NPI logistic heads and personnel who helped us in the field. We would also like to thank Chris Borstad for estimating flowlines to the rumple, Harvey Goodwin for assessing field safety, Vikram Goel for helping collect the data in field, and Robert Graham for providing the ERA5 data. Figures 1 and 2 were prepared using Quantarctica (quantarctica.npolar.no). For the REMA data set we acknowledge the following: DEMs provided by the Byrd Polar and Climate Research Center and the Polar Geospatial Center under NSF-OPP awards 1543501, 1810976, 1542736, 1559691, 1043681, 1541332, 0753663, 1548562, 1238993 and NASA award NNX10AN61G. Computer time provided through a Blue Waters Innovation Initiative. DEMs produced using data from DigitalGlobe, Inc.

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
