# Peer review of "Spatial and temporal variations in basal melting at Nivlisen ice"

_The Cryosphere, 2019_

## Referee Comment (RC1) · Anonymous Referee #1 · 17 Jun 2019

**Review: Spatial and temporal variations in basal melting at Nivlisen ice shelf, East Antarctica, derived from phase-sensitive radars**

This paper presents new measurements of sub-shelf melt rates of Nivlisen Ice Shelf in Dronning Maud Land, acquired with ApRES. The survey includes measurements across a broad area of the shelf at yearly resolution and at two points with 36-hour resolution, allowing the authors to study both spatial and temporal variations in melt. The melt rates on Nivlisen are found to be relatively modest, with the highest melt rates in the summer and just behind an ice rumple. These melt rate measurements are compared to a common-offset radar survey of ice-shelf thickness and to atmospheric data. While there is no correlation between ice-shelf thickness and melt rates, the atmospheric data suggest that the highest melt rates may be caused by wind pushing warm surface waters beneath the shelf.

The acquisition of ApRES data to determine melt rates is highly valuable as it allows direct measurement of ice-thickness changes while removing assumptions about firn thickness, strain rates, and/or hydrostatic equilibrium that affect other techniques. The authors have done a careful job in processing the data and assessing the uncertainty in the measurements, and performed extensive and detailed analysis of those results. Relatively few studies have used pRES on ice shelves, and the precision, temporal resolution, and relatively large spatial extent of these measurements make this paper a valuable insight into processes controlling melt, particularly beneath East Antarctic ice shelves. I have a number of comments, primarily focusing on the presentation and discussion, but I think the paper is a nice contribution and will soon be suitable for publication in *The Cryosphere*.

**General comments:**

The lengthy discussion of Jacobs et al.'s melt modes is too meandering to be easily followed. If this section is retained, I would recommend restructuring to introduce all 3 melt modes with brief definitions first before going on to detail each. In the conclusion, where mode-2 is mentioned again after having been absent since the introduction, it needs redefining. However, I do not find this division of modes to a very clean distinction for the purposes of this study. Instead, perhaps simply say that melt can be driven either by warm summer water near the surface or by warm water at depth and provide citations for each.

It seems like a stretch to call 4 m/yr "high" melt given the rates observed in West Antarctica. Sometimes this melt is described as "high" and sometimes as "higher"—I think remaining consistent calling it "higher" would be most clear.

The distinction between high melt and melt that is in excess of steady state gets a bit muddled here, partly because of the repeated use of the phrase "mass loss" to mean an outgoing flux of ice rather than a loss of total ice volume. I would suggest other terminology, such as "outgoing flux" or something similar, so as to clearly distinguish from a net loss. While I can figure out what is intended, I find the phrasing particularly distracting in the discussion of ice-shelf stability, since the measurements all indicate the melt rates at a particular, with no clear measure of whether those rates are sustainable or "normal". This ambiguity extends into the conclusions—most of the second paragraph of the conclusions is not a conclusion of this work, but more-or-less a hypothesis that "mode-3" melt may affect the stability of some ice shelves. It

is fine/good to make this argument, but I would not consider it a conclusion of this work and would find this paragraph more appropriate merged into the last section of the discussion (and perhaps reiterated in a single sentence in the conclusion).

In section 5.2, it would be nice to see a bit more connection between the different paragraphs. There is a lot of nice, detailed analysis of the phases and spectral power of the melt, but it is hard to know what to make of it in the aggregate. At present, the summary paragraph at the end of this section really just focuses on wind; it would be a huge help to use this paragraph to explain how the phase lead/lag of the seaward/landward sites can be related to the wind forcing, and to whether the spectral power of the melt at each site individually tells us anything about the validity of these conclusions.

All figures except Figure 1 should be enlarged. Simply expanding them to take up the full-page width would help significantly. Even with that expansion, though, some text needs to be further enlarged.

**Specific Comments:**

L41: For Nivlisen, surface melt/sublimation must be included in the inputs and outputs

L49:  Even though Rignot et al. state something similar this, I think this mischaracterizes the results of those studies; they both show calving and melt are equal within error.

L56: This sentence needs the context that this is the mode affecting the largest shelves

L75: What do you mean by "only recently"? Is this a change in occurrence or in observability? Why does this recentness suggest that it is important?

L121-124: Is the inland geography relevant anywhere in the rest of the paper? I think this can be removed

L129: Maybe move ice rise/rumple definition to where they are introduced in L114.

L160: Would be clearer to say "the ice front retreated to its present position by ~11 kyr ago"

L162-165: The wording here makes the meaning unclear—is the entrainment in line 163 the same as in 165, or are two different processes being described? In line 163, the reader needs to know what the CDW is being entrained into.

L198: Maybe mention the battery capacity here, since I'm sure others are considering similar deployments

L237-239: I'm not entirely clear what is meant here. You assume that strain varies either on very long timescales or on timescales shorter than 36H but not in between—essentially a bandstop filter? Are variations with the frequency of other tidal components small?

L271: Do you mean that the effect of horizontal positioning on the error in the vertical is 0.1±0.2 m?

L278: Citation for CSRS-PPP? Static or kinematic processing?

L299-301: I'm guessing you exclude the sites near the ice rumple because you were unable to revisit them? Perhaps mention this explicitly here.

L340: Can you say definitively that Bedmap2 is too high or could the thickness have changed?

L362-364: This sentence seems a bit backwards to me, but I know little about vorticity waves—can you clarify the mechanism for reducing melt rates and restructure the sentence so that cause and effect are clear?

L370-372: The language here should be made clearer. The measurements seem to indicate near perfect balance, so why would anything happen as a result of these rates being sustained?

L559: Based on the evidence provided in the paper, it would be more appropriate to say that the melt rates are susceptible rather than that the ice shelves are susceptible.

Figure 2: The color scales should be changed to match between the point measurements and the rasters in b-d.

Supplementary figure 1: Why is the x-axis in panel a in meters after a Fourier transform? Should it not be in Hz, or is this not the transformed data?

**Technical Corrections:**
L36: shrinking suggests extent, thinning would be more appropriate
L68: Tottem => Totten
L98: subject/verb disagreement
L103: "to explain them using" is an awkward phrase here
L154-155: This sentence needs a subject
L253: Line spacing of 5 km? Trace spacing of 5 m? I think there is a typo here.
L296: close to => just upstream of?
L305: average rate of thickness change
L561: there is a typo somewhere in "may increase leading"
L566: The first comma should not be there

---

## Referee Comment (RC2) · Anonymous Referee #2 · 25 Jun 2019

In this manuscript, the authors use an exciting ApRES data set to investigate basal melt rates underneath the Nivlisen ice shelf, East Antarctica. While repeat measurements of 29 ApRES sites distributed across and along ice-flow direction result in only 'moderate (0.8 m/yr)' annual basal melt rates, continuous records from two ApRES sites reveal a seasonal signal with 'highest daily' basal melt rates of up to 5.6 m/yr near the ice front. This seasonal signal cannot be observed at the second continuous ApRES site further upstream, which leads the authors to conclude that the presence of warm ocean surface water in summer and its interplay with the dominant winds in the area is the cause for the increased melt, rather than the intrusion of circumpolar deep water that causes very high basal melt rates in other parts of Antarctica. The authors support

their hypothesis with three GPR profiles, atmospheric data from both a nearby AWS and re-analysis data; and attempt the link of ApRES data to satellite imagery from MODIS.

In my opinion, the ApRES data set and the consequent quantification of basal melt rates in this area is required by the community to evaluate and improve current modelling efforts and I would very much like to see the manuscript published soon. The processing of the ApRES data is methodologically sound which makes this manuscript a valueable contribution to the study of ice-ocean interaction around Antarctica. I particularly enjoyed the thorough phase analysis between the two continuous ApRES records to display the seasonality in basal melting. The manuscript is mostly well organized but: (1) some parts of the extensive discussion can be shortened and belong to the description of the study area. Similarly, the writing style can be improved in many places. (2) The link to satellite data that is even underlined in the conclusion is weak which doesn't align with the author's very elegant analysis of ApRES data. (3) Some statements about the present pinning-points and their effect on ice-shelf stability can't be made with the data set presented. I recommend the manuscript for publication after minor revisions that include a revisit to the last part of the discussion section. I'm looking very much forward to it.

Minor comments:

l. 29-30: Including a statement about pinning points and their stabilizing effect made me anticipate a corresponding analysis in the main text. Without this analysis the statement is a bit too speculative to be included in the abstract. Reword

l. 35-36: I think with 'shrinking' you mean 'thinning'. I suggest changing to '...thinning glaciers in West Antarctica that lost back-stresses from their buttressing ice shelves."

l. 40-41: Change 'input of grounded ice upstream' to 'from ice across the grounding line' as it is a flux-gate calculation at the boundary between floating and grounded ice. Include 'underneath the floating ice shelf' after 'ocean' and 'at the ice front' after

'calving'. Also surface mass balance can be negative and represent ice loss. Please include in this list.

l. 43: Change to '...stresses on grounded ice upstream, leading the tributaries to flow faster' as there are more than one stress component to provide buttressing.

l. 45: Change to 'therefore the key to gain a...'

l. 52,62,75: I like the review of Jacobs melt modes and its link to basal melting around Antarctica. However I had to read these three paragraphs twice to follow. Reword to 'In mode 1,...' then 'In mode 2,...' and 'In mode 3,...' each followed by examples from the literature to help the reader. How about the high melt rates that have been observed in basal channels and lake drainage on Roi Baudouin or underneath the Whillans Ice Stream ? Please include in this review section.

l. 68: Change to 'Totten'

l. 81-82: This is hard to read. Change to '...reflect the integrated response to changes in circumpolar deep water temperatures and coastal processes that control its access onto the continental shelf (Thompson et al., 2018)' and please remove 'and the local upper ocean heat supply' as it doesn't add anything to the sentence. l. 99: Change 'resolution' to 'accuracy' or do you really mean vertical spatial resolution here ? Also change 'over' to 'and'

l. 104: Change 'explain' to 'interpret' as you only analyse the data at this section of the paper.

l. 106: Change 'were' to 'are'. General convention is to use past tense for everything that was done and present tense for everything that you have found out.

l. 107: Change to 'complement' as your data is plural

l. 108: Change to 'data source'

l. 112-113: Remove the first sentence as it doesn't add to the paper.

l. 117: Change to 'Basal melt rates from satellite data in...' to avoid the long concatenation

l. 121-124: Remove '100 km...ponds.' as this is trivia in the context of the paper.

l. 125-126: Change to '... has an estimated potential of raising global sea level by 8 cm.'

l. 132: you haven't introduced/defined the grounding zone yet. What do you mean exactly or can 'in the grounding zone' be removed ? For me a grounding zone is caused by tidal variability of ice mechanics downstream of the grounding line where ice detaches from the bed and becomes afloat.

l. 136: Change 'the shelf' to 'its stability'

l. 146: Include 'gradients' or 'heterogeneity' after 'surface mass balance'

l. 148: again 'in the grounding zone'

l. 157: Change '...remaining 25% coming from...' to '...residual 25% attributed to...' to avoid colloquial language

l. 158-165: This would be very interesting to see in your Fig. 1B (see specific comment below)

l. 170: Remove 'summertime' and change 'minimum to 'minima' as you also you 'maxima' earlier

l. 172: Reword 'dominant modes' as you introduced Jacobs modes earlier and you don't want to confuse the reader with additional modes

l. 177: Change 'then' to 'consequent'

l. 178: Change to 'remains'

l. 181-183: Include 'the' before 'Antarctic' and 'end'. The sentence about logistical support can be removed (you have it in the Acknowledgements already)

l. 185-186: Remove 'Below,...melt rates' as it doesn't add to the paper

l. 187: Change 'studied' to 'measured'

l. 189: Include 'all 29' after 'measured at' and change 'stake locations' to 'ApRES sites'

l. 190: Change to 'Autonomous phase-sensitive Radio Echo Sounder'

l. 191: Change 'speed' to 'velocity' as you mention the calculation of strain rates which require a direction. Velocity is speed with direction, speed doesn't have a direction. l. 193: Change 'shelf' to 'flow'

l. 195-196: 'Ice tongue' is this a common expression for this particular part of the ice shelf ? For me an ice tongue is a glacier that sticks out into the ocean without lateral thinning (for example the Drygalski Ice Tongue) and not a part of the floating ice shelf that is pushed through two ice rises like the one here.

l. 200 and elsewhere: 'stake sites' is confusing. Please reword throughout the paper

l. 217: Remove one of the two 'that'

l. 223-226: Reword this very long sentence. Also the word 'both' is used two times (the first one refers to actually three nouns). Maybe break it up into two sentences.

l. 233: Change 'returns' to 'reflector' and start a new sentence after 'processing' with 'This allowed us to. . .'

l. 236: The 36 h window size needs explanation.

l. 241: Include 'also' after 'we'. Sounds like 2016 was a busy field season !

l. 242: Remove 'across...structure' and replace with '...measurements on Nivlisen ice shelf (profiles A,B and C in Fig. 1b) as you have mentioned the orientation of the profiles already.

l. 246: there are three times the word 'with' in one line. Please reword
l. 248: Replace 'traces' with 'measurements'. Is 'code-phase' GPS special and improves your accuracy ? If it isn't I suggest removing it

l. 260-262: This sounds strange. Why is there such a big difference between the two methods to determine firn depth ? Also why is this important ? Did you use a 2-layer velocity model to convert travel time to depth ? I assume not. How did you determine 50 m firn from the ApRES data you present in Fig. S1 ? Please add some information here.

l. 262-263: Please add a sentence why the calculation of ice draft is necessary in this context. Also, for your freeboard calculation you require a sea level right ? Where does this come from ? A geoid model ?

l. 274: Remove 'We...Nivlisen.' as it doesn't add to the paper and is mentioned in Data and Methods section already

l.281: Change 'speed' to 'velocity'

l. 294: Again 'melt rates at stake locations'. Please reword

l. 294,296,297: It's called 'average annual'

l. 299-300: Reword and start the sentence with 'In 2018' to conform with the start of the paragraph

l. 304: 'low strain rates' compared to what ? Please add

l. 314: somewhere around here you move from using 'basal melt rates' to only 'melt rates'. Please remain consistent

l. 315: Include 'as' after the comma

l.316 and elsewhere: your 14 moth record ends in 2018 and not in 2017. Please change here and also in Figure captions.

l. 461-473: Most of this belongs to Section 2 Study Area where you explain the oceano-

graphic setting. Please move this paragraph, but still discuss earlier studies in a 'this confirms/is against the findings of way" at this stage.

l. 503-504: Same here, move to Section 2

l. 511-517: This is a nice paragraph and should also discuss potential links to Steward et al., 2019. Is this the same mechanism at play ?

l. 521: Change 'Fig. 7d' to 'Fig 7c'

l. 531-532: This statement needs to be defended with the right figure ! I suggest to change Fig. 7 (see below)

l. 533: Reword to '...was pushed by wind under the front of Nivlisen ice shelf'

l.534-539: I would swap these two sentences and begin with 'Surface wind' then say something about 'Surface warming' to get the order of processes right. End this paragraph here and remove the last sentence 'Natural...sea ice' as this more general statement that doesn't really fit here and creates an impression that actually weakens your results.

l. 548,551: Add values (0.8 and 5.6 m/yr) in braces after 'moderate' and 'summer'. Also add 'relatively' before 'high melt rates' as 5.6 m/yr are not high melt rates when I think of the Amundsen Sea.

l. 549: 'Daily' ? As far as I thought the temporal resolution of the data is much higher. More information is required on how you acquired the continuous ApRES data. Number of bursts/averaging/etc

l. 558: Change 'of' to 'in'

l. 559: Include 'temporally' before 'higher'. Also be consistent with 'basal melt rates' as it is called here 'rates of melting'

l. 564-565: Again 'pinning points'. I don't think that there is enough analysis on their

stability and how this might be affected by your measurements to include a statement like this in the conclusion. Please reword or move this to the discussion.

l. 570-571: Change 'important' to 'crucial' and remove 'which in turn is important for ice sheet models' as understanding the driving mechanism is much more important than including it into a model. By removing the last bit you put more emphasis on this.

Specific comments throughout the paper:

1. hyphenations in compound expressions are sometimes wrong or missing. For example l.131 'ice-shelf flow'. Hyphenation is wrong if no noun follows: 'the ice shelf flows' versus 'the ice-shelf flow'

2. 'Stake locations' I know that this comes from locating the ApRES antennas in the field over several years but somehow it sounds like you measure basal melting with stakes only. Can you reword 'Stake locations' to 'ApRES sites' and mention stakes only where you use them for the GPS survey and strain calculation ?

Figures: Figures are all way to small (see individual comments below)

Fig. 1) (a) what is the gray shaded area in the lower right ? (b) The ice-shelf front and the Landsat mosaic don't match up. Why is approx 1/3 of the ice shelf missing ? I suggest replacing the Landsat part of the figure with a schematic of what you know about the bathymetry (ridges, troughs, continental shelf edge) and the dominant oceanographic currents as you describe nicely in the main text (l. 158-165). Where was the carbon dating site ? Maybe remove the 'Ice structure' as you don't refer to them in the analysis of profile A-Aprime. (caption) Change 'made' to 'located'

Fig. 2) (a) colorbar for REMA DEM is missing, I like the absolute values of basal melt rates. (b) plot the difference of your GPR measurements to the Bedmap2 product and replace the colorbar with the new values. The contours stay the same, but you can tell where they match and where they don't. (c) similar here, color-code the stake sites with the difference to Measures and annotate the absolute measured value of Ice flow

[Figure]

velocity. (d) Same here, I'd display the difference in the markers and write the absolute measured SMB next to the stake sites. (caption) remove 'hill shade'

Fig. 3) Font size is incredibly small! First remove all repeated text from each of the three subplots. Each of the individual panels of the subplots use the same Distance so you only need to display that at the lower panel. The x-axis label 'Distance (km)' only needs to go below the third subplot. Also, all three surface elevation panels should have the same yaxis limits to be comparable. The radargram in the middle misses the blue surface elevation curve.

Fig. 4) (a) the start of the gray box c doesn't match with the start of your third subplot. (b) what do the white shaded areas in lower left and right mean ? (c) good (d) You don't need to write 'Time' when it is clear from the xaxis ticklabels. Maybe change 'Time' to '2017' (caption) the first 2017 is a 2018, right ? Ylabels 'melt rate' versus 'basal melt rate' earlier, pick one.

Fig. 5) (a) Consider writing '2017' and '2018' left and right next to the gray bars. (b and c) good (d) This looks like a spring-neap tidal signal over 14 days. Xticklabels should be the same as for Fig 4d. Consider replacing 'Time' with '2017'

Fig. 6) Very nice plot ! Don't use the same colormap as for Figs 4b and 5b as this is a different variable. Consider including a Legend with the arrow directions and 'in phase', 'seawards leads' and 'landward leads'. What do arrows pointing left stand for ? Also, has there been a threshold in coherence when you display the arrows ? What are the shaded areas in lower left and right ?

Fig. 7) I think this plot doesn't really show what you say in the main text. Both the temperature and sea-ice cover subplots didn't really help my understanding and could be moved to the supplements. Also, the interpretation of using dashed lines is to subjective to say that satellite data can't capture high melt events. I suggest: (I) using the space of subplots c and d and replace with a scatterplot of summertime wind speeds vs basal melt rates on the seaward site, where the dots are color-coded to wind direction

(similar to Fig. S5). (II) shade areas in (a) when you see open water in satellite data. Has the time lag between peaks in wind and basal melt rate only been estimated from the dashed lines ? That's ok, but it must be stated in the main text. (caption) Include 'nearby' before 'weather station'

Fig. S1) I can't see how a firn depth of 50m is derived from this plot, where does it come from and why is this important ? Change xaxis label to 'Depth below surface (m)'

Fig. S2) Comparing (c) to (d) indicates that there was less melt in 2018.

Fig. S3) (a) yaxis label is missing (b) include two xaxis labels '2017' and '2018'

Fig. S4) good

Fig. S5) (caption) Change '2017' to '2018'

Fig. S6) can you include the information about open water availability in your analysis ?

---

## Author Comment (AC1) · 19 Jul 2019

Dear Reviewer,

On behalf of all the authors of this discussion paper, I would like to thank you for your comments. Your suggestions have been acknowledged and have improved the paper substantially. Our responses can be found below.

Kind regards,

Katrin Lindbäck

RC1 General comments

[Figure]

RC1.1

This paper presents new measurements of sub-shelf melt rates of Nivlisen Ice Shelf in Dronning Maud Land, acquired with ApRES. The survey includes measurements across a broad area of the shelf at yearly resolution and at two points with 36-hour resolution, allowing the authors to study both spatial and temporal variations in melt. The melt rates on Nivlisen are found to be relatively modest, with the highest melt rates in the summer and just behind an ice rumple. These melt rate measurements are compared to a common-offset radar survey of ice-shelf thickness and to atmospheric data. While there is no correlation between ice-shelf thickness and melt rates, the atmospheric data suggest that the highest melt rates may be caused by wind pushing warm surface waters beneath the shelf.

The acquisition of ApRES data to determine melt rates is highly valuable as it allows direct measurement of ice-thickness changes while removing assumptions about firn thickness, strain rates, and/or hydrostatic equilibrium that affect other techniques. The authors have done a careful job in processing the data and assessing the uncertainty in the measurements, and performed extensive and detailed analysis of those results. Relatively few studies have used pRES on ice shelves, and the precision, temporal resolution, and relatively large spatial extent of these measurements make this paper a valuable insight into processes controlling melt, particularly beneath East Antarctic ice shelves. I have a number of comments, primarily focusing on the presentation and discussion, but I think the paper is a nice contribution and will soon be suitable for publication in The Cryosphere.

Author response:

Thanks for your positive comments, very much appreciated!

RC1.2

The lengthy discussion of Jacobs et al.'s melt modes is too meandering to be easily

followed. If this section is retained, I would recommend restructuring to introduce all 3 melt modes with brief definitions first before going on to detail each. In the conclusion, where mode-2 is mentioned again after having been absent since the introduction, it needs redefining. However, I do not find this division of modes to a very clean distinction for the purposes of this study. Instead, perhaps simply say that melt can be driven either by warm summer water near the surface or by warm water at depth and provide citations for each.

Author response:

We have restructured the section about the melt modes, clearly starting with each mode and its definition.

RC1.3

It seems like a stretch to call 4 m/yr "high" melt given the rates observed in West Antarctica. Sometimes this melt is described as "high" and sometimes as "higher"- I think remaining consistent calling it "higher" would be most clear.

Author response:

We have changed "high melt" to "higher melt" throughout the manuscript.

RC1.4

The distinction between high melt and melt that is in excess of steady state gets a bit muddled here, partly because of the repeated use of the phrase "mass loss" to mean an outgoing flux of ice rather than a loss of total ice volume. I would suggest other terminology, such as "outgoing flux" or something similar, so as to clearly distinguish from a net loss. While I can figure out what is intended, I find the phrasing particularly distracting in the discussion of iceshelf stability, since the measurements all indicate the melt rates at a particular, with no clear measure of whether those rates are sustainable or "normal". This ambiguity extends into the conclusions—most of the second paragraph of the conclusions is not a conclusion of this work, but more-or-less a hypothesis that "mode-3" melt may affect the stability of some ice shelves. It is fine/good to make this argument, but I would not consider it a conclusion of this work and would find this paragraph more appropriate merged into the last section of the discussion (and perhaps reiterated in a single sentence in the conclusion).

Author response:

We have changed the term "mass loss" to "stability". We have also moved the paragraph in the conclusions to the discussion. We hope is clearer now the distinction about the current status and potential future change.

RC1.5

In section 5.2, it would be nice to see a bit more connection between the different paragraphs. There is a lot of nice, detailed analysis of the phases and spectral power of the melt, but it is hard to know what to make of it in the aggregate. At present, the summary paragraph at the end of this section really just focuses on wind; it would be a huge help to use this paragraph to explain how the phase lead/lag of the seaward/landward sites can be related to the wind forcing, and to whether the spectral power of the melt at each site individually tells us anything about the validity of these conclusions.

Author response:

We have rewritten the last paragraph in the discussion give a better overview.

RC1.6

All figures except Figure 1 should be enlarged. Simply expanding them to take up the full-page width would help significantly. Even with that expansion, though, some text needs to be further enlarged.

Author response:

The figure size is fixed to a certain width by TC, if we understood it correctly, and cannot be set by us to fill the full-page width in a pdf. We have, nevertheless, enlarged the text

in the figures to make it more readable and will do a final check before publication to make sure it looks ok.

RC2 Specific comments

RC2.1

L41: For Nivlisen, surface melt/sublimation must be included in the inputs and outputs

Author response:

We have added surface melting and sublimation to the sentence.

RC2.2

L49: Even though Rignot et al. state something similar this, I think this mischaracterizes the results of those studies; they both show calving and melt are equal within error.

Author response:

We have rephrased the sentence as suggested (melt and calving equal).

RC2.3

L56: This sentence needs the context that this is the mode affecting the largest shelves

Author response:

We have clarified that this affects the largest ice shelves.

RC2.4

L75: What do you mean by "only recently"? Is this a change in occurrence or in observability? Why does this recentness suggest that it is important?

Author response:

We have removed "only recently".

RC2.5

L121-124: Is the inland geography relevant anywhere in the rest of the paper? I think this can be removed

Author response:

We have removed the sentences.

RC2.6

L129: Maybe move ice rise/rumple definition to where they are introduced in L114.

Author response:

We have moved the definition to the suggested section.

RC2.7

L160: Would be clearer to say "the ice front retreated to its present position by ∼11 kyr ago"

Author response:

We have rephrased the sentence as suggested.

RC2.8

L162-165: The wording here makes the meaning unclear - is the entrainment in line 163 the same as in 165, or are two different processes being described? In line 163, the reader needs to know what the CDW is being entrained into.

Author response:

The water is entrained into the Antarctic slope current. We have clarified the sentence.

RC2.9

L198: Maybe mention the battery capacity here, since I'm sure others are considering

similar deployments

Author response:

We have included the battery capacity.

RC2.10

L237-239: I'm not entirely clear what is meant here. You assume that strain varies either on very long timescales or on timescales shorter than 36H but not in between—essentially a bandstop filter? Are variations with the frequency of other tidal components small?

Author response:

This is not actually a bandstop filter. To get the mean melt rate, we needed to remove the time-average strain rate, which we needed to calculate elsewhere, essentially by comparing vertical profiles throughout the time series to see how the internal reflectors move with respect to each other. That correction sets the level of the melt rate. After that we assume that the main remaining vertical strain signal that needs to be removed is from tidal variation in the semi-diurnal and diurnal bands. So instead of trying to calculate the vertical strain rate at tidal frequencies (very difficult to do precisely because of the weakness of the internal reflections) we throw out all tidal variability (melt and strain) by filtering at 36 hours. That leaves us with the variability of most interest here. The assumption is that there is no significant tidal strain at frequencies slower than diurnal, except for the constant background strain rate. In some large ice shelves, a fortnightly signal is visible in the vertical strain rates, as a result of non-linear interactions between the diurnal and/or semidiurnal tides. That signal was not strong at these sites. We have clarified this in the text.

RC2.11

L271: Do you mean that the effect of horizontal positioning on the error in the vertical is 0.1±0.2 m?

Author response:

Yes, we have clarified this in the text.

RC2.12

L278: Citation for CSRS-PPP? Static or kinematic processing?

Author response:

We have added a reference to the processing and stated that it is static processing.

RC2.13

L299-301: I'm guessing you exclude the sites near the ice rumple because you were unable to revisit them? Perhaps mention this explicitly here.

Author response:

We have added a sentence clarifying this as suggested.

RC2.14

L340: Can you say definitively that Bedmap2 is too high or could the thickness have changed?

Author response:

Bedmap2 is 50-100 m off in this area and indicates an ice-rise like feature that cannot be seen in neither our radar data, our GNSS data nor the new REMA product. The ice-shelf is very flat in this area and a major change over the last few decades is unlikely. We edited the related sentence in the paper as follows: "The broad thickness pattern agrees with the gridded ice thickness of Bedmap2 (Fretwell et al., 2013), except on the western ice tongue (profile C), where the thickness of Bedmap2 is clearly too high (Fig. 2b), possibly due to errors in the input data or the interpolation between them."

RC2.15

L362-364: This sentence seems a bit backwards to me, but I know little about vorticity waves - can you clarify the mechanism for reducing melt rates and restructure the sentence so that cause and effect are clear?

Author response:

We have rephrased the sentence and left out the term "vorticity waves" and just described the strong tidal currents in shallow regions (thin water column thickness) around the ice rise that may increase the ice-ocean heat exchange.

RC2.16

L370-372: The language here should be made clearer. The measurements seem to indicate near perfect balance, so why would anything happen as a result of these rates being sustained?

Author response:

We have rephrased the sentence and replaced "sustained high melt rates" with "an increased basal melting in the future".

RC2.17

L559: Based on the evidence provided in the paper, it would be more appropriate to say that the melt rates are susceptible rather than that the ice shelves are susceptible.

Author response:

We have removed the word "susceptible" in the sentence.

RC2.18:

Figure 2: The color scales should be changed to match between the point measurements and the rasters in b-d.

Author response:

We have updated the figure as suggested by Reviwer #2, with difference between the in situ measured values and satellite or modelled values. The measured values are kept as numbers.

RC2.19

Supplementary Figure 1: Why is the x-axis in panel a in meters after a Fourier transform? Should it not be in Hz, or is this not the transformed data?

Author response:

For a FMCW radar, the frequency of each component of the data that are acquired represents the range to a reflector via the formula $R = T \cdot f \cdot v_i / (2 \cdot B)$, where $v_i$ is the radar speed in ice, $f$ is the frequency associated with the reflection at range $R$, $T$ is the length of the chirp in seconds, and $B$ is the bandwidth of the chirp. So we have taken the Fourier Transform, and converted to range using the above formula. We have clarified this in the figure caption.

RC3 Technical corrections

L36: shrinking suggests extent, thinning would be more appropriate

L68: Tottem => Totten

L98: subject/verb disagreement

L103: "to explain them using" is an awkward phrase here

L154-155: This sentence needs a subject

L253: Line spacing of 5 km? Trace spacing of 5 m? I think there is a typo here.

L296: close to => just upstream of?

L305: average rate of thickness change

L561: there is a typo somewhere in "may increase leading"

L566: The first comma should not be there Author response

Author response:

We have corrected all these errors. Thanks for pointing them out!

---

## Author Comment (AC2) · 19 Jul 2019

Dear Reviewer,

On behalf of all the authors of this discussion paper, I would like to thank you for your comments. Your suggestions have been acknowledged and have improved the paper substantially. Our responses can be found below.

Kind regards,

Katrin Lindbäck

RC1 General comments

[Figure]

In this manuscript, the authors use an exciting ApRES data set to investigate basal melt rates underneath the Nivlisen ice shelf, East Antarctica. While repeat measurements of 29 ApRES sites distributed across and along ice-flow direction result in only 'moderate (0.8 m/yr)' annual basal melt rates, continuous records from two ApRES sites reveal a seasonal signal with 'highest daily' basal melt rates of up to 5.6 m/yr near the ice front. This seasonal signal cannot be observed at the second continuous ApRES site further upstream, which leads the authors to conclude that the presence of warm ocean surface water in summer and its interplay with the dominant winds in the area is the cause for the increased melt, rather than the intrusion of circumpolar deep water that causes very high basal melt rates in other parts of Antarctica. The authors support their hypothesis with three GPR profiles, atmospheric data from both a nearby AWS and re-analysis data; and attempt the link of ApRES data to satellite imagery from MODIS.

In my opinion, the ApRES data set and the consequent quantification of basal melt rates in this area is required by the community to evaluate and improve current modelling efforts and I would very much like to see the manuscript published soon. The processing of the ApRES data is methodologically sound which makes this manuscript a valuable contribution to the study of ice-ocean interaction around Antarctica. I particularly enjoyed the thorough phase analysis between the two continuous ApRES records to display the seasonality in basal melting. The manuscript is mostly well organized but: (1) some parts of the extensive discussion can be shortened and belong to the description of the study area. Similarly, the writing style can be improved in many places. (2) The link to satellite data that is even underlined in the conclusion is weak which doesn't align with the author's very elegant analysis of ApRES data. (3) Some statements about the present pinning-points and their effect on ice-shelf stability can't be made with the data set presented. I recommend the manuscript for publication after minor revisions that include a revisit to the last part of the discussion section. I'm looking very much forward to it.

Author response:

We are very grateful for your positive review. We have taken into account all your main suggestions and detailed responses can be found below.

RC2 Minor comments

RC2.1

l. 29-30: Including a statement about pinning points and their stabilizing effect made me anticipate a corresponding analysis in the main text. Without this analysis the statement is a bit too speculative to be included in the abstract. Reword

Author response:

We have removed the sentence in the abstract.

RC2.2

l. 35-36: I think with 'shrinking' you mean 'thinning'. I suggest changing to '...thinning glaciers in West Antarctica that lost back-stresses from their buttressing ice shelves."

Author response:

We have changed the wording as suggested.

RC2.3

l. 40-41: Change 'input of grounded ice upstream' to 'from ice across the grounding line' as it is a flux-gate calculation at the boundary between floating and grounded ice. Include 'underneath the floating ice shelf' after 'ocean' and 'at the ice front' after 'calving'. Also surface mass balance can be negative and represent ice loss. Please include in this list.

Author response:

We have rewritten the sentence as suggested.
RC2.4

l. 43: Change to '...stresses on grounded ice upstream, leading the tributaries to flow faster' as there are more than one stress component to provide buttressing.

Author response:

We have rewritten the sentence as suggested.

RC2.5

l. 45: Change to 'therefore the key to gain a: : :'

Author response:

We have rewritten the sentence as suggested.

RC2.6

l. 52,62,75: I like the review of Jacobs melt modes and its link to basal melting around Antarctica. However I had to read these three paragraphs twice to follow. Reword to 'In mode 1,: : :' then 'In mode 2,: : :' and 'In mode 3,: : :' each followed by examples from the literature to help the reader. How about the high melt rates that have been observed in basal channels and lake drainage on Roi Baudouin or underneath the Whillans Ice Stream ? Please include in this review section.

Author response:

We have restructured the section as suggested. We have added a reference to Whillans Ice Stream about high melting at basal channels. We did not include surface melting from Roi Baudouin, but it is reviewed in the Study area section

RC2.7

l. 68: Change to 'Totten'

Author response:

We have corrected the typo.

RC2.8

l. 81-82: This is hard to read. Change to '...reflect the integrated response to changes in circumpolar deep water temperatures and coastal processes that control its access onto the continental shelf (Thompson et al., 2018)' and please remove 'and the local upper ocean heat supply' as it doesn't add anything to the sentence.

Author response:

We have changed the sentence as suggested.

RC2.9

l. 99: Change 'resolution' to 'accuracy' or do you really mean vertical spatial resolution here ? Also change 'over' to 'and'

Author response:

We have changed the words as suggested.

RC2.10

l. 104: Change 'explain' to 'interpret' as you only analyse the data at this section of the paper.

Author response:

We have rephrased the sentence as suggested.

RC2.11

l. 106: Change 'were' to 'are'. General convention is to use past tense for everything that was done and present tense for everything that you have found out.

Author response:

We have changed the tense.

RC2.12

l. 107: Change to 'complement' as your data is plural

Author response:

We have corrected the typo.

RC2.13

l. 108: Change to 'data source'

Author response:

We have changed as suggested.

RC2.14

l. 112-113: Remove the first sentence as it doesn't add to the paper.

Author response:

We have removed the sentence.

RC2.15

l. 117: Change to 'Basal melt rates from satellite data in: : :' to avoid the long concatenation

Author response:

We have changed the sentence.

RC2.16

l. 121-124: Remove '100 km...ponds.' as this is trivia in the context of the paper.

Author response:

We have removed the section.

RC2.17

l. 125-126: Change to ': : : has an estimated potential of raising global sea level by 8cm.'

Author response:

We have changed the sentence.

RC2.18

l. 132: you haven't introduced/defined the grounding zone yet. What do you mean exactly or can 'in the grounding zone' be removed ? For me a grounding zone is caused by tidal variability of ice mechanics downstream of the grounding line where ice detaches from the bed and becomes afloat.

Author response:

We have added a definition of the grounding line/zone, where the term is first introduced.

RC2.19

l. 136: Change 'the shelf' to 'its stability'

Author response:

We have changed as suggested.

RC2.20

l. 146: Include 'gradients' or 'heterogeneity' after 'surface mass balance'

Author response:

We have added the word 'gradients'.

RC2.21

l. 148: again 'in the grounding zone'

Author response:

We have added a definition of the grounding zone earlier in the text.

RC2.22

l. 157: Change '...remaining 25% coming from...' to '...residual 25% attributed to: : :' to avoid colloquial language

Author response:

We have changed the sentence.

RC2.23

l. 158-165: This would be very interesting to see in your Fig. 1B (see specific comment below)

Author response:

We have not included the carbon dating sites in the figure, since there were several of them and they were not part of this study.

RC2.24

l. 170: Remove 'summertime' and change 'minimum to 'minima' as you also you 'maxima' earlier

Author response:

We have corrected the word.

RC2.25

l. 172: Reword 'dominant modes' as you introduced Jacobs modes earlier and you

don't want to confuse the reader with additional modes

Author response:

We changed the word 'modes' to 'trends'.

RC2.26

l. 177: Change 'then' to 'consequent'

Author response:

We have changed the word.

RC2.27

l. 178: Change to 'remains'

Author response:

We have corrected the typo.

RC2.28

l. 181-183: Include 'the' before 'Antarctic' and 'end'. The sentence about logistical support can be removed (you have it in the Acknowledgements already)

Author response:

We have corrected the text, but would like to keep the station description in the text. We have removed the reference in the Acknowledgements.

RC2.29

l. 185-186: Remove 'Below,...melt rates' as it doesn't add to the paper

Author response:

We have removed the sentences.
RC2.30

l. 187: Change 'studied' to 'measured'

Author response:

We have changed the word.

RC2.31

l. 189: Include 'all 29' after 'measured at' and change 'stake locations' to 'ApRES sites'

Author response:

We have changed the wording.

RC2.32

l. 190: Change to 'Autonomous phase-sensitive Radio Echo Sounder'

Author response:

We have changed the heading.

RC2.33

l. 191: Change 'speed' to 'velocity' as you mention the calculation of strain rates which require a direction. Velocity is speed with direction, speed doesn't have a direction. l.

Author response:

We have changed as suggested.

RC2.34

l. 193: Change 'shelf' to 'flow'

Author response:

We have changed the word.

RC2.35

l. 195-196: 'Ice tongue' is this a common expression for this particular part of the ice shelf ? For me an ice tongue is a glacier that sticks out into the ocean without lateral thinning (for example the Drygalski Ice Tongue) and not a part of the floating ice shelf that is pushed through two ice rises like the one here.

Author response:

We have removed 'ice tongue' from the manuscript.

RC2.36

l. 200 and elsewhere: 'stake sites' is confusing. Please reword throughout the paper

Author response:

We have changed 'stake sites' to 'ApRES sites'.

RC2.37

l. 217: Remove one of the two 'that'

Author response:

We have corrected the typo.

RC2.38

l. 223-226: Reword this very long sentence. Also the word 'both' is used two times (the first one refers to actually three nouns). Maybe break it up into two sentences.

Author response:

We have rewritten the sentence into two.

RC2.39

l. 233: Change 'returns' to 'reflector' and start a new sentence after 'processing' with

'This allowed us to: : :'

Author response:

We have changed the sentence as suggested.

RC2.40

l. 236: The 36 h window size needs explanation.

Author response:

To get the mean melt rate, we needed to remove the time-average strain rate, which we needed to calculate elsewhere, essentially by comparing vertical profiles throughout the time series to see how the internal reflectors move with respect to each other. That correction sets the level of the melt rate. After that we assume that the main remaining vertical strain signal that needs to be removed is from tidal variation in the semi-diurnal and diurnal bands. So instead of trying to calculate the vertical strain rate at tidal frequencies (very difficult to do precisely because of the weakness of the internal reflections) we throw out all tidal variability (melt and strain) by filtering at 36 hours. That leaves us with the variability of most interest here. The assumption is that there is no significant tidal strain at frequencies slower than diurnal, except for the constant background strain rate. In some large ice shelves, a fortnightly signal is visible in the vertical strain rates, as a result of non-linear interactions between the diurnal and/or semidiurnal tides. That signal was not strong at these sites. We have clarified this in the text.

RC2.41

l. 241: Include 'also' after 'we'. Sounds like 2016 was a busy field season !

Author response:

We have included the word as suggested.

RC2.42

l. 242: Remove 'across...structure' and replace with '...measurements on Nivlisen ice shelf (profiles A,B and C in Fig. 1b) as you have mentioned the orientation of the profiles already.

Author response:

We have changed the sentence as suggested.

RC2.43

l. 246: there are three times the word 'with' in one line. Please reword

Author response:

We have rephrased the sentence as suggested.

RC2.44

l. 248: Replace 'traces' with 'measurements'. Is 'code-phase' GPS special and improves your accuracy ? If it isn't I suggest removing it

Author response:

We have changed the words, however, we have kept the 'code-phase' description since it is a different GPS than the 'carrier-phase', which has better accuracy.

RC2.45

l. 260-262: This sounds strange. Why is there such a big difference between the two methods to determine firn depth ? Also why is this important ? Did you use a 2-layer velocity model to convert travel time to depth ? I assume not. How did you determine 50 m firn from the ApRES data you present in Fig. S1 ? Please add some information here.

Author response:

We added a plot to Figure S1, where we show how the assumption of 50 m firn depth was made. We did not use a 2-layer velocity model, since it was not necessary for the purpose of this study, and the density error is included in the uncertainty number.

RC2.46

l. 262-263: Please add a sentence why the calculation of ice draft is necessary in this context. Also, for your freeboard calculation you require a sea level right ? Where does this come from ? A geoid model ?

Author response:

We have added a sentence why ice draft is important and added information about the geoid.

RC2.47

l. 274: Remove 'We...Nivlisen.' as it doesn't add to the paper and is mentioned in Data and Methods section already

Author response:

We have removed the sentence.

RC2.48

l.281: Change 'speed' to 'velocity'

Author response:

We have changed the word.

RC2.49

l. 294: Again 'melt rates at stake locations'. Please reword

Author response:

We have changed it to 'ApRES sites'.

RC2.50

l. 294,296,297: It's called 'average annual'

Author response:

We have changed it to 'averaged annual' throughout the manuscript.

RC2.51

l. 299-300: Reword and start the sentence with 'In 2018' to conform with the start of the paragraph

Author response:

We have reworded the sentence.

RC2.52

l. 304: 'low strain rates' compared to what ? Please add

Author response:

We have removed the statement.

RC2.53

l. 314: somewhere around here you move from using 'basal melt rates' to only 'melt rates'. Please remain consistent

Author response:

We have added 'basal' to 'melt rates' in many places throughout the manuscript.

RC2.54

l. 315: Include 'as' after the comma

Author response:

We have corrected the sentence as suggested.

RC2.55

l.316 and elsewhere: your 14 moth record ends in 2018 and not in 2017. Please change here and also in Figure captions.

Author response:

We have corrected the typo, here and in the figure captions. Thanks for noticing.

RC2.56

l. 461-473: Most of this belongs to Section 2 Study Area where you explain the oceano-graphic setting. Please move this paragraph, but still discuss earlier studies in a 'this confirms/is against the findings of way" at this stage.

Author response:

We have moved the paragraph to the Study area.

RC2.57

l. 503-504: Same here, move to Section 2

Author response:

We have rephrased the sentence. The statement is mentioned in the Introduction.

RC2.58

l. 511-517: This is a nice paragraph and should also discuss potential links to Steward et al., 2019. Is this the same mechanism at play ?

Author response:

We have added a paragraph comparing with Stewart et al. (2019). In similarity with

our study they find a link to solar-heated surface water, but they did not find any link to downwelling-favourable winds.

RC2.59

l. 521: Change 'Fig. 7d' to 'Fig 7c'

Author response:

We have corrected the figure number.

RC2.60

l. 531-532: This statement needs to be defended with the right figure ! I suggest to change Fig. 7 (see below)

Author response:

We have updated Fig. 7 and also added a figure (Fig. 8) from the Supplements to support this statement.

RC2.61

l. 533: Reword to '...was pushed by wind under the front of Nivlisen ice shelf'

Author response:

We have changed the sentence as suggested.

RC2.62

l.534-539: I would swap these two sentences and begin with 'Surface wind' then say something about 'Surface warming' to get the order of processes right. End this paragraph here and remove the last sentence 'Natural...sea ice' as this more general statement that doesn't really fit here and creates an impression that actually weakens your results.

Author response:

We have rearranged the sentences and removed the last sentence as suggested.

RC2.63

l. 548,551: Add values (0.8 and 5.6 m/yr) in braces after 'moderate' and 'summer'. Also add 'relatively' before 'high melt rates' as 5.6 m/yr are not high melt rates when I think of the Amundsen Sea.

Author response:

We have added the numbers to the conclusions.

RC2.64

l. 549: 'Daily' ? As far as I thought the temporal resolution of the data is much higher. More information is required on how you acquired the continuous ApRES data. Number of bursts/averaging/etc

Author response:

We have changed 'daily' to 'hourly'.

RC2.65

l. 558: Change 'of' to 'in'

Author response:

We have changed the word.

RC2.66

l. 559: Include 'temporally' before 'higher'. Also be consistent with 'basal melt rates' as it is called here 'rates of melting'

Author response:

We have changed as suggested.

RC2.67

l. 564-565: Again 'pinning points'. I don't think that there is enough analysis on their stability and how this might be affected by your measurements to include a statement like this in the conclusion. Please reword or move this to the discussion.

Author response:

We have moved this section to the discussion.

RC2.68

l. 570-571: Change 'important' to 'crucial' and remove 'which in turn is important for ice sheet models' as understanding the driving mechanism is much more important than including it into a model. By removing the last bit you put more emphasis on this.

Author response:

We have changed the sentence as suggested.

RC3 Specific comments throughout the paper

RC3.1

1. hyphenations in compound expressions are sometimes wrong or missing. For example l.131 'ice-shelf flow'. Hyphenation is wrong if no noun follows: 'the ice shelf flows' versus 'the ice-shelf flow'

Author response:

We have corrected this at several places and will check this in detail once more for the final version of the paper.

RC3.2

2. 'Stake locations' I know that this comes from locating the ApRES antennas in the field over several years but somehow it sounds like you measure basal melting with

stakes only. Can you reword 'Stake locations' to 'ApRES sites' and mention stakes only where you use them for the GPS survey and strain calculation ?

Author response:

We have changed 'stake locations' to 'ApRES sites'.

RC3.3

Figures: Figures are all way to small (see individual comments below)

Author response:

Figures are not allowed to have full width in the pdf-version of the paper. We have increased the size of the text and it is possible to click and zoom in the figures online. We will check this again in detail before the final version.

RC3.4

Fig. 1) (a) what is the gray shaded area in the lower right ? (b) The ice-shelf front and the Landsat mosaic don't match up. Why is approx 1/3 of the ice shelf missing? I suggest replacing the Landsat part of the figure with a schematic of what you know about the bathymetry (ridges, troughs, continental shelf edge) and the dominant oceanographic currents as you describe nicely in the main text (l. 158-165). Where was the carbon dating site ? Maybe remove the 'Ice structure' as you don't refer to them in the analysis of profile A-Aprime. (caption) Change 'made' to 'located'

Author response:

The figure has been updated with elevation contours to show bathymetric ridges and the continental shelf edge. The ice-shelf front is outlined with a contour as described in the legend, were the Landsat image also shows sea ice north of the ice front. We have clarified this in the figure caption. We would like to keep the ice structure, since it shows ice-shelf characteristics. We have not added the carbon dating sites, since there were many sites and they were not part of this study. We have removed the grey

shaded area in Fig. 1a. We have reworded the figure caption as suggested.

RC3.5

Fig. 2) (a) colorbar for REMA DEM is missing, I like the absolute values of basal melt rates. (b) plot the difference of your GPR measurements to the Bedmap2 product and replace the colorbar with the new values. The contours stay the same, but you can tell where they match and where they don't. (c) similar here, color-code the stake sites with the difference to Measures and annotate the absolute measured value of Ice flow velocity. (d) Same here, I'd display the difference in the markers and write the absolute measured SMB next to the stake sites. (caption) remove 'hill shade'

Author response:

We have updated the figure as suggested with difference between the in situ measured values and satellite or modelled. We removed the REMA hillshade in Fig. 2a since it was difficult to interpret.

RC3.6

Fig. 3) Font size is incredibly small! First remove all repeated text from each of the three subplots. Each of the individual panels of the subplots use the same Distance so you only need to display that at the lower panel. The x-axis label 'Distance (km)' only needs to go below the third subplot. Also, all three surface elevation panels should have the same yaxis limits to be comparable. The radargram in the middle misses the blue surface elevation curve.

Author response:

We have updated the figure as suggested. We would like to keep the distance as it is for Profile B and C, since then it is possible to see details like the basal channels, which would not be possible if the profile was the same scale as the very long Profile A.

RC3.7
Fig. 4) (a) the start of the gray box c doesn't match with the start of your third subplot. (b) what do the white shaded areas in lower left and right mean ? (c) good (d) You don't need to write 'Time' when it is clear from the xaxis ticklabels. Maybe change 'Time' to '2017' (caption) the first 2017 is a 2018, right ? Ylabels 'melt rate' versus 'basal melt rate' earlier, pick one.

Author response:

We have improved the figures as suggested. The grey shaded areas in the lower left and right are the cone of influence, where edge effects become important and the image can be distorted. We have clarified this in the figure caption

RC3.8

Fig. 5) (a) Consider writing '2017' and '2018' left and right next to the gray bars. (b and c) good (d) This looks like a spring-neap tidal signal over 14 days. Xticklabels should be the same as for Fig 4d. Consider replacing 'Time' with '2017'

Author response:

We have improved the figures as suggested.

RC3.9

Fig. 6) Very nice plot ! Don't use the same colormap as for Figs 4b and 5b as this is a different variable. Consider including a Legend with the arrow directions and 'in phase', 'seawards leads' and 'landward leads'. What do arrows pointing left stand for ? Also, has there been a threshold in coherence when you display the arrows ? What are the shaded areas in lower left and right ?

Author response:

Thanks! We have changed the colour map and inserted a legend with the arrows as suggested. Within the cone of influence, shown as a lighter shade, edge effects become important and the image can be distorted. We have clarified this in the figure

caption.

RC3.10

Fig. 7) I think this plot doesn't really show what you say in the main text. Both the temperature and sea-ice cover subplots didn't really help my understanding and could be moved to the supplements. Also, the interpretation of using dashed lines is to subjective to say that satellite data can't capture high melt events. I suggest: (I) using the space of subplots c and d and replace with a scatterplot of summertime wind speeds vs basal melt rates on the seaward site, where the dots are color-coded to wind direction (similar to Fig. S5). (II) shade areas in (a) when you see open water in satellite data. Has the time lag between peaks in wind and basal melt rate only been estimated from the dashed lines ? That's ok, but it must be stated in the main text. (caption) Include 'nearby' before 'weather station'

Author response:

We have updated the figure as suggested and added a shaded grey area for the time period of open water. We added an updated version of Fig. S5 from the supplements as Fig. 8. We have clarified in the text that the dashed lines are where the time lags have been calculated.

RC3.11

Fig. S1) I can't see how a firn depth of 50m is derived from this plot, where does it come from and why is this important ? Change xaxis label to 'Depth below surface (m)'

Author response:

We have added a subplot with the residual height to show the assumption about the firn depth. We changed Depth to Range.

RC3.12

Fig. S2) Comparing (c) to (d) indicates that there was less melt in 2018.
Author response:

We have added a sentence in the caption from the results: "Basal melt rates were slightly lower in the second year at 18 sites and for 8 sites slightly higher."

RC3.13

Fig. S3) (a) yaxis label is missing (b) include two xaxis labels '2017' and '2018'

Author response:

We have adjusted the labels as suggested.

RC3.14

Fig. S4) good

Author response:

Thanks!

RC3.15

Fig. S5) (caption) Change '2017' to '2018'

Author response:

The figure was moved to the manuscript as Figure 8. The figure shows scatter plot between overlapping time periods when there was open water (11 Dec 2016-1 Mar 2017).

RC3.16

Fig. S6) can you include the information about open water availability in your analysis?

Author response:

We have included this in the Discussion section.